# Towards Learning to Complete Anything in Lidar

**Ayça Takmaz**[† 1 2]   **Cristiano Saltori**[1]   **Neehar Peri**[1 3]   **Tim Meinhardt**[1]
**Riccardo de Lutio**[1]   **Laura Leal-Taixé**[1]   **Aljosa Osep**[1]

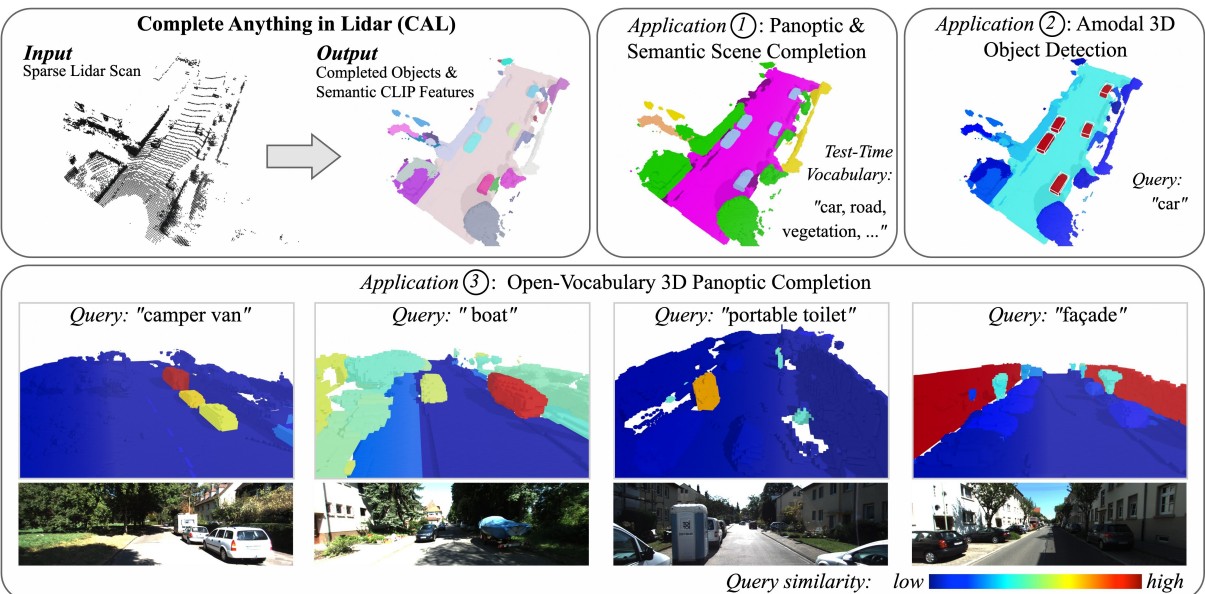

*Figure 1.* **Learning to Complete Anything in Lidar**. Given a sparse Lidar point cloud, **CAL** (*Complete Anything in Lidar*) localizes, reconstructs, and, optionally, recognizes objects in a zero-shot fashion. By providing a semantic class vocabulary of specific object classes *at test time*, **CAL** can be prompted to perform Semantic Scene Completion (SSC), Panoptic Scene Completion (PSC), or (amodal) 3D Object Detection. *Note that* **CAL** *only takes a single Lidar scan as input; RGB images are shown for visualization purposes only.*

## Abstract

We propose **CAL** (*Complete Anything in Lidar*) for Lidar-based shape-completion in-the-wild. This is closely related to Lidar-based semantic/panoptic scene completion. However, contemporary methods can only complete and recognize objects from a closed vocabulary labeled in existing Lidar datasets. Different to that, our zero-shot approach leverages the temporal context from multi-modal sensor sequences to mine object shapes and semantic features of observed objects. These are then distilled into a Lidar-only instance-level completion and recognition model. Although we only mine partial shape completions, we find that our distilled model learns to infer full object shapes from multiple such partial observations across the dataset. We show that our model can be prompted on standard benchmarks for Semantic and Panoptic Scene Completion, localize objects as (amodal) 3D bounding boxes, and recognize objects beyond fixed class vocabularies.

## 1. Introduction

Understanding the complete spatial layout and semantics of objects and scene geometry from raw sensor data is crucial for embodied 3D perception and safe navigation.

Contemporary methods for Lidar semantic (Behley et al., 2019) and instance (Behley et al., 2021) segmentation only group and classify points directly observable from the Lidar sensor. In contrast, methods for (amodal) 3D object detection (Zhou & Tuzel, 2018), Semantic Scene Completion (SSC) (Behley et al., 2019; Li et al., 2024) and Panoptic Scene Completion (PSC) (Cao et al., 2024) learn to complete objects and scenes directly from labeled sensor data to predict occluded regions not directly observable in Lidar. However, prior work can only localize and complete around

[†]Work done during a research internship at NVIDIA. [1]NVIDIA [2]ETH Zurich [3]Carnegie Mellon University (CMU). Correspondence to: Ayça Takmaz <ayca.takmaz@inf.ethz.ch>.

20 classes labeled in existing Lidar datasets. This is *far* below the label diversity and scale compared to state-of-the-art image-based datasets (Kirillov et al., 2023).

**Mining shape priors from unlabeled data.** Unlike prior work, we extend beyond typical fixed taxonomies by learning object shape priors from temporal context in Lidar sequences. However, this requires (i) segmenting objects and regions in space *and* time, followed by (ii) temporal aggregation to obtain 3D shape estimates. Precisely tracking objects is crucial for 3D reconstruction: tracking errors like identity switches can lead to erroneous or incomplete geometry. Even when objects are correctly localized in space and time, object-level temporal registration is challenging (Groß et al., 2019; Seidenschwarz et al., 2024; Huang et al., 2022).

To address these challenges, we leverage image (Kirillov et al., 2023) and video (Ravi et al., 2024) segmentation foundation models to localize and track objects in video. Such foundation models are *already* trained on diverse data and are capable of accurately segmenting *any* object in a video. After segmenting and tracking all objects, we lift each masklet (i.e. spatio-temporal object masks) to Lidar space and integrate them over time using a calibrated multi-modal sensor setup with known ego-vehicle poses (Behley et al., 2019). To enable zero-shot recognition, we additionally obtain CLIP (Radford et al., 2021) features for each masklet averaged per-timestep, effectively connecting the completed shapes with semantic information. Objects may be only partially completed, and not all objects are static – however, in practice, we learn to fully complete static and dynamic objects from such partial observations.

**Learning to complete anything.** We utilize these mined pseudo-label pairs to train an instance-level completion network. Following prior methods for SSC (Behley et al., 2019), we learn scene-level occupancy in a fixed-size voxel grid using a sparse generative encoder-decoder network (Cao et al., 2024). To segment individual object instances, we train a decoder (Cao et al., 2024), which predicts instance masks in the (occupied) voxel space in a class-agnostic fashion. Finally, we regress a CLIP token (Najibi et al., 2023) for each predicted instance to capture object semantics in a fixed-dimensional embedding vector, enabling zero-shot prompting at test time.

We empirically demonstrate that our method is versatile and can be used for semantic (Behley et al., 2019; Li et al., 2024), and panoptic scene completion (Cao et al., 2024) (Fig. 1, ①) via test-time prompts. Moreover, we qualitatively show that our approach can localize objects as 3D bounding boxes (Fig. 1, ②) and demonstrate that our method can recognize and complete arbitrary objects not captured in canonical semantic vocabularies (Fig. 1, ③).

**Contributions.** We propose the first method for Zero-Shot Lidar Panoptic Scene Completion. Our approach is enabled by our pseudo-labeling engine, which mines 3D shape priors from unlabeled Lidar sequences using 2D vision foundation models. We show that such pseudo-labels can be used to train a model for scene-scale object-level completion from noisy and partial pseudo-labels.

## 2. Related Work

In the following section, we discuss prior work with a focus on Lidar-based scene understanding, scene completion, and generative modeling.

**Lidar-based segmentation** methods for semantic (Behley et al., 2019) and panoptic (Behley et al., 2021) segmentation classify *directly* observed Lidar points into pre-defined semantic classes and identify individual instances. While most existing methods rely on manually labeled datasets (Behley et al., 2019; 2021; Fong et al., 2021), several works (Unal et al., 2022; Li et al., 2023a) address weakly supervised segmentation to reduce labeling efforts. More recent works embrace foundation models for open-vocabulary auto-labeling. Liu et al. (2023) use contrastive pre-training to distill vision-foundation model features for label-efficient segmentation. Osep et al. (2024) distill vision foundation models into a zero-shot Lidar panoptic segmentation model. Peng et al. (2023); Xiao et al. (2024) similarly distill 2D foundational knowledge into 3D but rely on Lidar *and* camera inputs to classify Lidar/RGB-D points at test-time. Such *modal* scene recognition only localizes the visible portion of objects and is therefore sensitive to signal sparsity and (self) occlusions. In contrast, our method addresses zero-shot completion of shapes in Lidar, which comes with distinct challenges beyond the scope of segmention.

**Lidar-based object detection** methods localize objects as oriented 3D bounding boxes (Petrovskaya & Thrun, 2009; Yin et al., 2021; Liu et al., 2021; Ma et al., 2023; Peri et al., 2023a;b), including regions not directly observed by the Lidar sensor. As these methods require manually labeled 3D boxes, recent work (Najibi et al., 2022; 2023; Zhang et al., 2023; Seidenschwarz et al., 2024; Khurana et al., 2024) utilize foundational priors and temporal context to automatically obtain amodal 3D boxes for *moving* objects. In contrast, our work is not limited to the subset of *thing* classes observed in a state of motion – we learn to segment and complete objects for *any* category.

**Object-level shape completion** from partially observed scans (*e.g.*, a single viewpoint) is commonly addressed using data-driven methods that rely on object shape priors (Yuan et al., 2018; Dai et al., 2017b; Mescheder et al., 2019; Park et al., 2019; Mittal et al., 2022). Prior art utilizes gener-

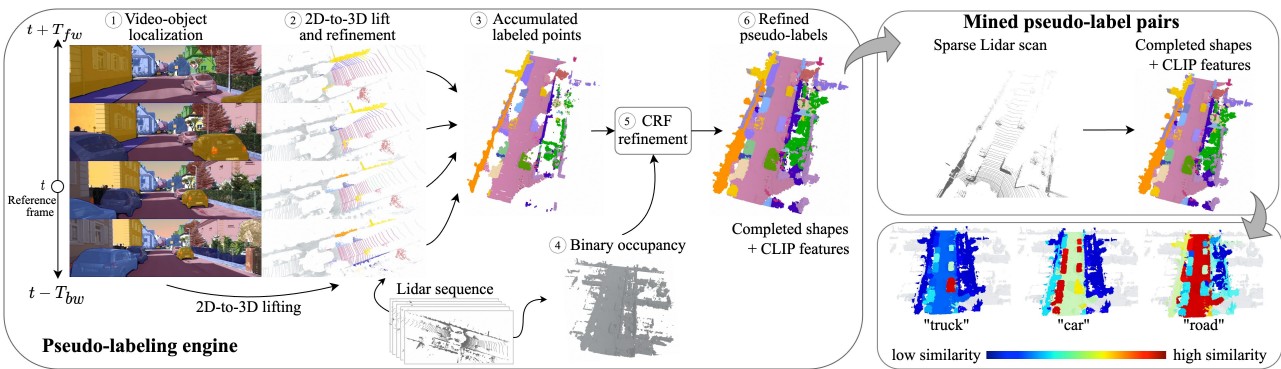

*Figure 2.* **Pseudo-labeling engine.** Given a calibrated RGB camera and Lidar sensor, ① we use video-object segmentation models (Ravi et al., 2024) to localize object instances in video, ② pseudo-label the Lidar point clouds over time, and ③ generate completed voxelized object representations, each enriched with a per-instance CLIP feature extracted from RGB images. In ④, we accumulate 360° Lidar scans to obtain full-scene binary occupancy, used for refining the aggregated pseudo-labels ③ via a CRF-guided label refinement process ⑤. As output ⑥, our method pairs each sparse and incomplete Lidar scan with pseudo-labels for *object-level* scene completion (top-right) and CLIP features, which are temporally aggregated by averaging per-instance features across the sequence. These CLIP features enable zero-shot recognition via text queries (bottom-right). Mined pseudo-label pairs are then used to train the `CAL` model.

ative models such as adversarial networks (Wu et al., 2016; Achlioptas et al., 2018), auto-encoders (Wu et al., 2015; Dai et al., 2017b; Yuan et al., 2018), diffusion models (Vahdat et al., 2022; Nam et al., 2022; Ntavelis et al., 2023; Jun & Nichol, 2023), and vector-quantized auto-encoders (Van Den Oord et al., 2017; Mittal et al., 2022). Such models are commonly trained on datasets that provide partial observations and corresponding 3D meshes (Chang et al., 2015; Deitke et al., 2023), or via data augmentations (Dai et al., 2017b; Mittal et al., 2021). In contrast, we address object *and* scene-level completion via auto-labeling.

**Scene-level completion** methods typically represent labeled, pre-registered scans of static environments using dense voxel grids, which hinders scalability to large scenes (Dai et al., 2017a; 2018). To address this, Dai et al. (2020) utilize sparse encoders in conjunction with coarse-to-fine generation in a feed-forward manner, while Ren et al. (2024) perform diffusion in the latent space. This differs from *streaming robot perception*, where range sensors provide sparse measurements (Song et al., 2017; Behley et al., 2019; Li et al., 2024) and methods must infer dense scene geometry in the presence of dynamic objects. Li et al. (2023b) explore implicit scene completion of sparse Lidar data, whereas Nunes et al. (2024) explore diffusion-based completion of Lidar point clouds. Prior work addressing *semantic* scene completion (Xia et al., 2023; Liang et al., 2024; Roldao et al., 2020; Rist et al., 2021; Mei et al., 2023) requires labeled datasets (which are expensive to curate) (Behley et al., 2019; Li et al., 2024; Tian et al., 2024) with amodal completion data.

Cao et al. (2024) jointly tackle SSC and instance segmentation (*i.e.*, PSC) using a generative encoder-decoder in conjunction with a transformer decoder that interacts with

occupied voxels to estimate a set of segmentation masks for each (learned) query vector. Although our model is inspired by Cao et al. (2024), our work goes beyond fixed class taxonomies, and does not require 3D datasets with manual labels. Our work addresses these limitations by extracting geometry from temporal cues and open-vocabulary semantics from foundational priors. Finally, open-vocabulary SSC from *multi-view images* has been explored in Vobecky et al. (2023); Zheng et al. (2024). To the best of our knowledge, no prior work combines open-vocabulary, *instance-level*, and *LiDAR-based* scene completion in a single framework.

## 3. Method

We describe our task formulation and method `CAL` (*Complete Anything in Lidar*) for zero-shot completion of objects in a sparse Lidar scan. `CAL` has two key components: (i) a pseudo-labeling engine (Fig. 2) that mines pairs of partially observed point clouds with completed 3D shapes and CLIP features (Radford et al., 2021), and (ii) a model for zero-shot, class-agnostic object completion (Fig. 3).

**Preliminary: Panoptic Scene Completion.** Semantic Scene Completion (SSC) (Behley et al., 2019) assumes input in the form of a *single* Lidar point cloud $P = \{p_n\}_{n=1}^{N}, p_n \in \mathbb{R}^4$, consisting of spatial positions and intensity channel. Given a set of semantic classes (known at train time and accompanied by labeled instances), the task is to estimate per-class scene occupancy. In addition, Panoptic Scene Completion (PSC) (Cao et al., 2024) requires assigning $K$ instance identities to semantic classes designated as *thing* classes. Object shapes are localized and parametrized via a per-object occupancy function $O_k : \mathbb{R}^4 \to \mathbb{N}^3, k \leq K$, defined in a regular voxel grid $\mathcal{G} \subset \mathbb{N}^3$ in a fixed-size bounding volume in front of the Lidar sensor.

**Complete Anything.** Our problem setting follows the general setting for PSC. However, particular to our setting, semantic information is *not* provided during training. Our method takes a semantic vocabulary consisting of free-form semantic class descriptions *only* at test time. We address this challenge by performing class-agnostic segmentation and reconstruction of objects, optionally followed by zero-shot classification, that assigns each instance to a semantic class in the specified test-time vocabulary.

### 3.1. Mining 3D Shape Priors From Unlabeled Data

**Overview.** Given a Lidar scan $P$ with partial scene observations, our shape-mining pseudo-labeler creates an occupancy grid $O$ where each voxel can be empty, unlabeled, or assigned to an observed instance. Each instance is represented by a 3D occupancy grid $O_k$, accumulating per-point instance observations over time. Additionally, each $O_k$ is accompanied by a semantic CLIP feature $f_k$, that connects instance geometry with vision-language features (Radford et al., 2021). We depict our approach for mining shape priors in Fig. 2.

**Key challenge.** Our approach stems from the observation that object shapes can be inferred from temporal context while driving down the street. However, obtaining *per-object* shape priors requires precise object localization, tracking and accurate object-level registration – a challenging problem sensitive to registration errors and drift (Huang et al., 2022; Seidenschwarz et al., 2024; Groß et al., 2019).

**Video-object localization** ①. To this end, we utilize segmentation foundation models (Kirillov et al., 2023; Ravi et al., 2024; Radford et al., 2021), trained on a large amount of diverse visual data and proven capable of segmenting arbitrary objects. Given a video sequence $v$ consisting of RGB images $I_t \in \mathbb{R}^{W \times H \times 3}$, we first segment images with SAM (Kirillov et al., 2023) using grid-based prompting and obtain a set of 2D mask proposals $m_{t,k}^{2D} \in \{0,1\}^{W \times H}, 0 \le k < K$, that represent the set of $\le K$ objects in the reference image, observed at time $t$. We propagate the masks from the reference frame at time $t$ to the frames within the given temporal window $[t - T_{bw}, t + T_{fw}]$ using video-object segmentation model SAM 2 (Ravi et al., 2024). Specifically, we perform *backward propagation* for $T_{bw}$ frames and *forward propagation* for $T_{fw}$ frames, with a stride of $w$. This allows us to integrate observations from various viewpoints. The output is a set of $\le K$ class-agnostic masklets, providing temporal instance association in the video $v$.

**Semantic feature aggregation.** We compute per-instance CLIP features following Ding et al. (2023), and aggregate (Takmaz et al., 2023) them in the temporal domain using mask information to obtain multi-view vision-language features. These features are then normalized and averaged to obtain $f_k \in \mathbb{R}^F$ for each accumulated shape $k$ in the reference frame $t$, where $F$ is the CLIP embedding dimension.

**Lift and refine** ②. For each frame in the temporal window, we backproject each 2D mask $m_{t,k}^{2D}$ to the Lidar coordinate frame using the provided camera-to-Lidar transformations and obtain 3D masks $m_{t,k}^{3D} \in \{0,1\}^{X \times Y \times Z}, 0 \le k < K$ for each instance $k$. These masks may suffer from projection artifacts due to imperfect calibration and rolling shutter. We follow the single-frame mask refinement procedure proposed by Osep et al. (2024) for precise localization (details in Appx. A.3).

**Temporal aggregation** ③,④. Once masks $m_{t,k}^{3D}$ are localized in the Lidar sequence, we project them into the reference coordinate frame using known ego-poses and aggregate them over time to obtain densified masks $m_k^{3D}$. We obtain per-instance occupancy $O_k$ by voxelizing each $m_k^{3D}$ and assign instance indices via majority voting. Relying solely on $O_k$ provides limited guidance. This is due to (i) image-based object localization providing only partial coverage of Lidar scans where unseen regions (e.g., the back of a car) lack completion signals and (ii) errors in mask localization and tracking. We complement $O_k$ with a binary occupancy signal obtained by directly accumulating $360°$ Lidar points (Fig. 2 ④) to improve label coverage (Tab. 5).

**CRF refinement** ⑤. To further improve label coverage, we refine instance masks ③ using Conditional Random Fields (CRF) (Krähenbühl & Koltun, 2011), constructed over binary occupancy ④. We analyze the benefits of this CRF refinement (whose final output is illustrated in Fig. 2 ⑥) in Tab. 3 and report further details in the Appx. A.1.

### 3.2. Learning To Complete Objects

**Overview.** `CAL` takes a single *input* Lidar scan $P$, providing sparse and incomplete observations of scene geometry (Fig. 4, *1st col.*), and *outputs* a set of completed object instances (Fig. 4, *2nd col.*), represented via voxel occupancy and instance IDs. Each instance ID is accompanied by a predicted semantic feature that allows us to match instance semantic features with the class vocabulary provided at test time (Fig. 4, *3rd col.*). Different from prior art (Cao et al., 2024), we do not differentiate between *stuff* and *thing* classes, *i.e.*, cars, road, and (individual) bushes are treated as individual instances. `CAL` overview is provided in Fig. 3.

**Model architecture.** We follow Cao et al. (2024) and employ a sparse-generative 3D U-Net (Dai et al., 2018) architecture that estimates scene-level occupancy, and a Transformer instance decoder (Cheng et al., 2022) operating directly on occupied voxels. The backbone consists of a sparse feature encoder (●) (Choy et al., 2019) followed by a dense 3D convolutional block (●). The multi-scale genera-

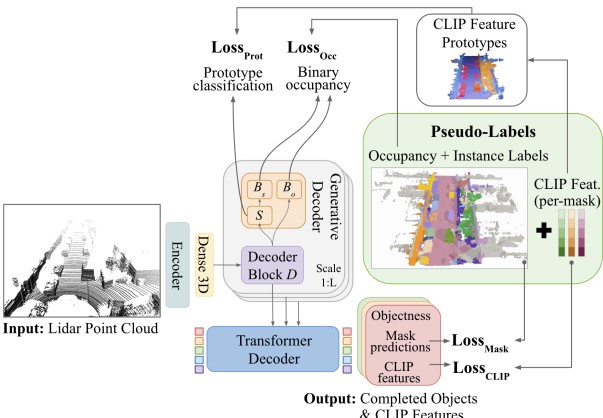

*Figure 3.* **CAL model architecture and training pipeline.** The backbone consists of a sparse encoder and a dense 3D convolutional block. We estimate scene-level occupancy using a multi-scale sparse generative decoder that consists of decoder blocks $D$, two occupancy heads $B_o$ and $B_s$, and a pseudo-semantic head ($S$) at each scale L. The Transformer decoder then predicts segmentation masks over the completed scene and regresses CLIP features.

tive decoder (⬤) uses three decoding blocks $D^{1:L}$ estimating occupancy at three different resolution levels $L \in \{1, 2, 4\}$. The input to the Transformer decoder (⬤) is a set of learnable queries that interact with the multi-resolution features learned by the generative decoder in voxel space. To accommodate zero-shot recognition, we estimate an objectness score, a segmentation mask (parametrized in the voxel space), and CLIP features (⬤) for each query.

**Generative decoder, completion heads & CLIP feature prototypes.** The key component for accurately *completing* the scene is the sparse generative decoder (Dai et al., 2018; Cao et al., 2024), comprising three decoder blocks interleaved with pruning layers to maintain sparsity. Each decoder block $D^{1:L}$ provides features at scale L, used by a binary occupancy head $B_o$ to predict scene occupancy at that scale. Training the current model with pseudo-labels presents two key challenges: partially completed pseudo-labels bias training toward well-covered regions, and the lack of a semantic grouping of instances hinders the learning of shape priors. We address the first issue by guiding $B_o$ with binary occupancy (Fig. 2 ④). We address the second challenge by quantizing instance CLIP features into $C$ pseudo-prototypes with clustering and introduce an additional (pseudo) semantic head $S$ to predict a prototype class for each occupied voxel. We use prototype predictions only during training (additional details in Appx. B.2) – during test-time, we only use the predicted CLIP features for zero-shot recognition. Additionally, we add a second occupancy head $B_s$, which processes per-class logits from $S$ and learns to convert them into a binary occupancy prediction. We find that $B_s$ further regularizes our training.

**Training.** We train our network jointly for (i) binary oc-

cupancy completion, (ii) class-agnostic instance mask prediction, (iii) CLIP feature distillation, and (iv) per-voxel prototype assignment. During each training iteration, the generative decoder produces coarse-to-fine voxel grids for each scale $L$, supervised with a binary occupancy loss ($\mathcal{L}_{occ}$: binary-cross entropy *wrt.* aggregated binary occupancy labels) and prototype classification loss ($\mathcal{L}_{prot}$: cross-entropy and Lovasz *wrt.* pseudo-category labels from CLIP prototype assignments). The transformer decoder produces instance masks and CLIP features, supervised by the mask-loss ($\mathcal{L}_{mask}$: binary-cross entropy and Dice loss) and the feature distillation loss ($\mathcal{L}_{CLIP}$: cosine similarity loss). For $\mathcal{L}_{mask}$, we perform Hungarian matching to pair predicted and pseudo-labeled instance masks. We *ignore* unlabeled voxels for Hungarian matching, and also for computing $\mathcal{L}_{mask}$, and $\mathcal{L}_{prot}$ to ensure the network is not penalized for predictions outside the pseudo-labeled area. The final training objective is the weighted sum of $\{\mathcal{L}_{occ}, \mathcal{L}_{prot}, \mathcal{L}_{CLIP}, \mathcal{L}_{mask}\}$ with weighting parameters specified as $\{\lambda_{occ}, \lambda_{prot}, \lambda_{CLIP}, \lambda_{mask}\}$. Further implementation details are provided in Appx. B.3.

**Inference.** Given a Lidar point cloud as input, **CAL** produces a set of object instance masks over the voxel grid and a CLIP feature for each predicted instance. We suppress small overlapping masks with the overlap threshold $\tau_{ovr}$ and filter low confidence masks with the objectness threshold $\tau_{obj}$ (Appx. B.2). We use the predicted CLIP features to classify each query in a zero-shot manner. Using the CLIP text encoder, we encode the semantic vocabulary, i.e., a set of text prompts (additional details in Appx. C.1). We then obtain a posterior over the specified vocabulary for each query by computing the cosine similarity between a predicted CLIP feature and encoded text descriptions.

## 4. Experiments

We evaluate **CAL**'s ability to localize and complete the full 3D extent of instances from a Lidar point cloud given a class vocabulary prompt at test time. Key details on the experimental setup (Sec. 4.1), benchmark comparisons (Sec. 4.2), and ablations on design choices (Sec. 4.3- 4.3) are discussed below, with further implementation details in the Appendix.

### 4.1. Experimental Setup

**Task.** We quantitatively assess **CAL**'s zero-shot completion and recognition performance on Semantic Scene Completion (SSC) (Behley et al., 2019) and Panoptic Scene Completion (PSC) (Cao et al., 2024) benchmarks. SSC assumes a single Lidar point cloud as input and requires per-voxel occupancy estimation and semantic classification of the occupied voxels. PSC additionally requires assigning identities to instances of *thing* classes. Given a class vocabulary

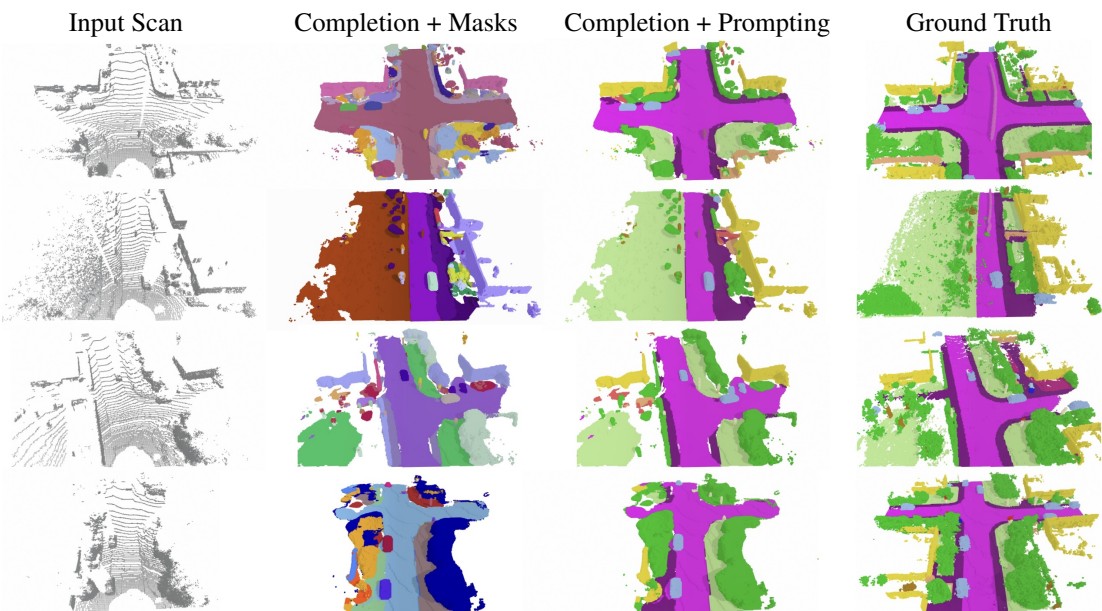

| Input Scan | Completion + Masks | Completion + Prompting | Ground Truth |

*Figure 4.* **Qualitative results on SemanticKITTI.** Given a single Lidar scan ($1^{st}$ col.), **CAL** completes object-level observations as a set of masks over the voxel grid ($2^{nd}$ col.) and predicts a CLIP feature for each mask. We can prompt with any semantic class vocabulary and obtain panoptic and semantic scene completion ($3^{rd}$ col.) results. Our model predicts shape priors for both *thing* (e.g., 'car', 'cyclist') and *stuff* classes (e.g. 'vegetation', 'road') and can correctly predict the intersection geometry in $4^{th}$ row, despite limited direct evidence.

*Table 1.* **Panoptic Scene Completion.** We compare **CAL** against LMSCNet (Roldao et al., 2020) + MaskPLS (Marcuzzi et al., 2023), JS3CNet (Yan et al., 2021) + MaskPLS, SCPNet (Xia et al., 2023) + MaskPLS, and PaSCo (Cao et al., 2024) (M=1 and Ensemble).

| | Semantic KITTI (Behley et al., 2019) (val set) | | | | | | | | | | | SSCBench-KITTI360 (Li et al., 2024) (test set) | | | | | | | | | | |
| | All | | | Thing | | | Stuff | | | | All | | | | Thing | | | Stuff | | | |
| Method | $PQ^{\dagger}\uparrow$ | $PQ\uparrow$ | SQ | RQ | PQ | SQ | RQ | PQ | SQ | RQ | mIoU↑ | $PQ^{\dagger}\uparrow$ | $PQ\uparrow$ | SQ | RQ | PQ | SQ | RQ | PQ | SQ | RQ | mIoU↑ |
|---|---|---|---|---|---|---|---|---|---|---|---|---|---|---|---|---|---|---|---|---|---|---|
| *Fully supervised* | | | | | | | | | | | | | | | | | | | | | | |
| LMSCNet + MaskPLS | 13.81 | 4.17 | 36.13 | 6.82 | 1.62 | 29.87 | 2.68 | 6.02 | 40.69 | 9.82 | 17.02 | 12.76 | 4.14 | 26.52 | 6.45 | 0.88 | 20.41 | 1.58 | 5.78 | 29.58 | 8.88 | 15.10 |
| JS3CNet + MaskPLS | 18.41 | 6.85 | 41.90 | 11.34 | 4.18 | 43.10 | 7.22 | 8.79 | 41.03 | 14.34 | 22.70 | 16.42 | 6.79 | 51.16 | 10.71 | 3.36 | 48.41 | 5.83 | 8.51 | 52.54 | 13.15 | 21.31 |
| SCPNet + MaskPLS | 19.39 | 8.59 | 49.49 | 13.69 | 4.88 | 46.41 | 7.70 | 11.30 | 51.73 | 18.04 | 22.44 | 16.54 | 6.14 | 51.18 | 10.15 | 4.23 | 48.46 | 7.05 | 7.09 | 52.55 | 11.70 | 21.47 |
| PaSCo (M=1) | 26.49 | 15.36 | 54.15 | 23.65 | 12.33 | 47.42 | 18.78 | 17.55 | 59.05 | 27.19 | 28.22 | 19.53 | 9.91 | 58.81 | 15.40 | 3.46 | 57.72 | 6.10 | 13.14 | 59.35 | 20.05 | 21.17 |
| PaSCo (Ensemble) | 31.42 | 16.51 | 54.25 | 25.13 | 13.71 | 48.07 | 20.68 | 18.54 | 58.74 | 28.38 | 30.11 | 26.29 | 10.92 | 56.10 | 17.09 | 4.88 | 57.53 | 8.48 | 13.94 | 55.39 | 21.39 | 22.39 |
| *Zero-shot* | | | | | | | | | | | | | | | | | | | | | | |
| **CAL** (SO) | 17.12 | 6.27 | 43.40 | 10.06 | 3.48 | 44.39 | 5.65 | 8.30 | 42.67 | 13.27 | 20.71 | 12.56 | 1.71 | 33.18 | 3.10 | 2.05 | 45.57 | 3.76 | 1.54 | 26.99 | 2.76 | 13.34 |
| **CAL** (ZS) | 13.12 | 5.26 | 27.45 | 8.44 | 2.42 | 22.79 | 3.89 | 7.33 | 30.84 | 11.76 | 13.09 | 8.57 | 1.46 | 21.01 | 2.63 | 1.39 | 27.62 | 2.54 | 1.49 | 17.81 | 2.68 | 8.49 |

for evaluation, we prompt our method at inference time to perform these tasks in a zero-shot manner.

**Datasets and benchmarks.** We follow prior work (Cao et al., 2024) and evaluate **CAL** on two datasets that provide semantic *and* instance-level labels for PSC: SSCBench-KITTI360 (Li et al., 2024; Liao et al., 2021) and SemanticKITTI (Behley et al., 2019; Geiger et al., 2012; 2013) whose instance-level labels are provided by Cao et al. (2024). The hyperparameters used by our pseudo-labeling engine for each dataset are given in Appx. A, and additional details are provided in Appx. C.1.

**Zero-shot evaluation.** We evaluate our model's segmentation, completion, and recognition capabilities by specifying target classes (defined in each respective dataset) via prompts at test time (additional details in Appx. C.1) and assign labels based on cosine similarity between encoded text prompts and the predicted CLIP features $f_k$. While

our model is trained to localize and classify a larger set of objects (*i.e.*, objects that appear in the Lidar data and are localized by vision-based segmentation foundation models, Kirillov et al. (2023)), we utilize labeled instances as a proxy for assessing zero-shot (ZS) shape completion and recognition on classes that are labeled. *In contrast to baselines trained on ground-truth (GT) data, we use GT labels solely for evaluation and ablations.*

**Metrics.** We assess SSC performance using mean Intersection-over-Union (mIoU). For PSC, we follow Cao et al. (2024) and use the Panoptic Quality ($PQ = SQ \times RQ$), Segmentation Quality ($SQ$), and Recognition Quality ($RQ$) metrics (Kirillov et al., 2019), evaluated on the full voxel grid. Similar to Cao et al. (2024), we focus on the modified variant of $PQ$, i.e. $PQ^{\dagger}$ (Behley et al., 2021), and remove the minimum $0.5$ IoU overlap requirement for *stuff* classes, as this can be too restrictive for regions that do not have well-defined boundaries. As $PQ$ mixes seg-

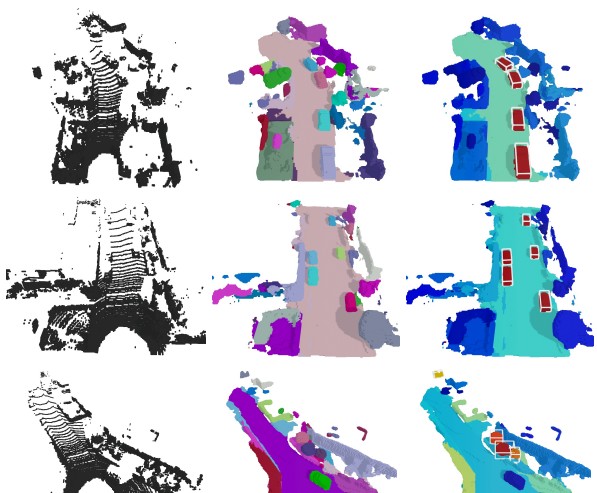

*Figure 5.* **Completion and amodal detection on KITTI-360.** Given an input Lidar scan (left), **CAL** outputs a set of completed object shapes (middle). We visualize recognized objects (right) for queries 'vehicle' (top), 'car' (middle) and 'tree' (bottom), and fit 3D bounding boxes to the identified object instances, demonstrating the zero-shot amodal 3D object detection ability of **CAL**.

mentation and recognition accuracy, we additionally report semantic oracle *(SO)* results (similar to Osep et al. (2024)) by assigning each mask to the GT semantic label based on a per-voxel majority vote. This allows us to further decouple completion performance and semantic understanding.

### 4.2. Experimental results

First, we compare **CAL** with four fully supervised baselines in Tab. 1. To the best of our knowledge, our method is the first method to address zero-shot Lidar panoptic scene completion; therefore, we also construct two zero-shot baselines, whose results are reported in Tab. 2. Additionally, we report qualitative results on SemanticKITTI in Fig. 4 and Fig. 6, and on SSCBench-KITTI360 in Fig. 5 and Appx. Fig. 7.

**Comparison to supervised baselines.** Tab. 1 reports **CAL** results for SSC and PSC on the SemanticKITTI and SSCBench-KITTI360 datasets. We also compare with four fully supervised baselines. We report two versions of the PaSCo baseline, the current state-of-the-art method for fully supervised SSC/PSC: single network ($M = 1$), whose settings are consistent with ours, and the ensembled version (Ensemble), reported only for completeness. Fully supervised baselines have a clear advantage over **CAL** as they train on closed-set, noise-free annotations with full scene coverage. Nonetheless, even early supervised SSC baselines (Behley et al., 2019) struggle, often achieving less than 15 mIoU[1], outlining the difficulty of the task even

---

[1]The SSC baselines reported in (Behley et al., 2019) achieve 9.53 mIoU (SSCNet (Song et al., 2017)), 9.54 mIoU (TS3D (Garbade et al., 2019)), and 17.70 mIoU (w/ SATNet (Liu et al., 2018))

*Table 2.* **Panoptic scene completion results with zero-shot baselines.** We compare **CAL** against the zero-shot baselines we construct: LODE (Li et al., 2023b) + SAL (Osep et al., 2024) and LiDiff (Nunes et al., 2024) + SAL (Osep et al., 2024). Results reported on the SemanticKITTI dataset.

| Method | All | | | | Thing | | | Stuff | | | SSC |
| --- | --- | --- | --- | --- | --- | --- | --- | --- | --- | --- | --- |
| | $PQ^{\dagger}\uparrow$ | $PQ\uparrow$ | SQ | RQ | PQ | SQ | RQ | PQ | SQ | RQ | mIoU↑ |
| LODE + SAL | 7.74 | 1.96 | 11.12 | 3.54 | 0.00 | 6.36 | 0.00 | 3.39 | 14.59 | 6.11 | 8.12 |
| LiDiff + SAL | 7.35 | 0.36 | 23.95 | 0.65 | 0.22 | 34.81 | 0.40 | 0.46 | 16.06 | 0.83 | 7.38 |
| **CAL** (ZS) | 13.12 | 5.26 | 27.45 | 8.44 | 2.42 | 22.79 | 3.89 | 7.33 | 30.84 | 11.76 | 13.09 |

in the supervised setting[1]. Remarkably, we reach $\sim 50\%$ of PaSCo on SemanticKITTI, and $\sim 40\%$ of PaSCo on SSCBench-KITTI360 in the zero-shot (ZS) setting, while even achieving comparable results to fully-supervised baseline LMSCNet + MaskPLS. Specifically, we achieve 13.12 $PQ^{\dagger}$ (49.51 % of PaSCo) and 13.09 mIoU (46.37 % of PaSCo) in the ZS setting on SemanticKITTI and further improve to 17.12 $PQ^{\dagger}$ (64.63 % of PaSCo) with the semantic oracle (SO). Similarly, **CAL** achieves 8.57 $PQ^{\dagger}$ (43.87 % of PaSCo) and 8.49 mIoU (40.11 % of PaSCo), narrowing the gap to 17.12 $PQ^{\dagger}$ (64.63 %) with SO on SSCBench-KITTI-360. We find that the gap between **CAL** and the supervised baselines is largely due to zero-shot recognition performance, which is limited by the underlying vision foundation model. In our per-class analysis (further details in Appx. C.3), we also see that the gap between **CAL** and the supervised baselines is affected by rare classes (e.g., pedestrian and cyclist).

**Comparison to zero-shot baselines.** As there are no prior works tackling Lidar PSC in zero-shot setting, we construct two baselines adhering to the following criteria for a fair zero-shot comparison: (1) input should be a *single Lidar scan*, (2) scene completion model should be trained *without semantic labels*, and (3) instance prediction and semantic inference should rely on *zero-shot* recognition. Accordingly, we combined recent Lidar completion methods that were trained *without semantic labels* – LODE (Li et al., 2023b) and LiDiff (Nunes et al., 2024) – with SAL (Osep et al., 2024), a zero-shot panoptic segmentation model.

LODE performs implicit scene completion from sparse Lidar, trained with ground-truth completion data. We employ the LODE variant that does not use any semantic labels. We extract a surface mesh from its output, convert it to an occupancy grid, and propagate SAL's zero-shot per-point panoptic labels to occupied voxels (LODE+SAL, Tab. 2). LiDiff is a diffusion-based completion method that learns to complete Lidar point clouds from GT completion data without semantic labels. Similarly, we convert its output to an occupancy grid and propagate SAL's zero-shot panoptic labels to voxels (LiDiff+SAL, Tab. 2).

As can be seen in Fig. 6 and Tab. 2, our method outperforms zero-shot baselines across nearly all metrics. Notably, while

| Input Scan | LiDiff + SAL | LODE + SAL | Ours (CAL) | Ground Truth |
|---|---|---|---|---|

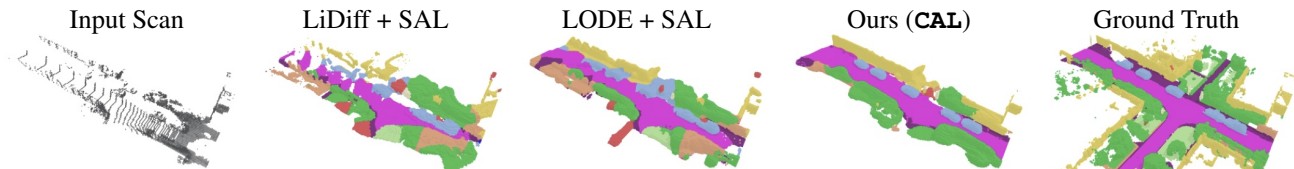

*Figure 6.* **Comparison to zero-shot baselines on SemanticKITTI.** Given a single Lidar scan ($1^{st}$ col.), we compare **CAL** ($4^{th}$ col.) to zero-shot baselines ($2^{nd}$ and $3^{rd}$ cols.) combining LiDiff (Nunes et al., 2024) and LODE (Li et al., 2023b) with SAL (Osep et al., 2024).

*Table 3.* **CRF refinement ablation.** We evaluate pseudo-label quality with and without CRF refinement on SemanticKITTI and SSCBench-KITTI360. Results show that CRF refinement significantly improves pseudo-label quality in both datasets and settings.

| | Semantic KITTI (Behley et al., 2019) (val set) | | | | | | | | | | | SSCBench-KITTI360 (Li et al., 2024) (test set) | | | | | | | | | | |
| | | All | | | Thing | | | Stuff | | | | | All | | | Thing | | | Stuff | | | |
| Method | $PQ^{\dagger}\uparrow$ | $PQ\uparrow$ | SQ | RQ | PQ | SQ | RQ | PQ | SQ | RQ | mIoU↑ | $PQ^{\dagger}\uparrow$ | $PQ\uparrow$ | SQ | RQ | PQ | SQ | RQ | PQ | SQ | RQ | mIoU↑ |
|---|---|---|---|---|---|---|---|---|---|---|---|---|---|---|---|---|---|---|---|---|---|---|
| *Pseudo-label* | | | | | | | | | | | | | | | | | | | | | | |
| **CAL** (SO) | 12.78 | 5.59 | 46.22 | 9.39 | 5.00 | 49.79 | 8.56 | 6.02 | 43.63 | 10.00 | 17.77 | 11.48 | 4.44 | 35.06 | 7.14 | 3.24 | 47.17 | 5.81 | 5.04 | 29.00 | 7.81 | 13.22 |
| **CAL** (ZS) | 9.34 | 3.45 | 35.07 | 5.83 | 1.75 | 48.14 | 3.08 | 4.69 | 25.56 | 7.83 | 12.79 | 9.37 | 3.66 | 28.80 | 5.82 | 1.67 | 47.23 | 3.00 | 4.66 | 19.58 | 7.22 | 9.72 |
| *Pseudo-label + CRF* | | | | | | | | | | | | | | | | | | | | | | |
| **CAL** (SO) | 25.90 | 17.71 | 64.06 | 26.55 | 16.93 | 67.91 | 23.75 | 18.28 | 61.25 | 28.58 | 33.10 | 14.75 | 4.57 | 40.79 | 7.98 | 7.08 | 48.18 | 12.35 | 3.32 | 37.10 | 5.79 | 17.14 |
| **CAL** (ZS) | 16.98 | 10.67 | 54.02 | 16.19 | 7.30 | 67.19 | 10.21 | 13.12 | 44.43 | 20.54 | 20.62 | 10.98 | 3.18 | 33.95 | 5.58 | 3.91 | 48.30 | 6.86 | 2.82 | 26.78 | 4.95 | 11.04 |

*Table 4.* **Pseudo-labeling engine ablations, semantic oracle (SO).** Pseudo-labels benefit from forward and backward propagation, with notable improvements up to $T_{fw} = 32$ and $T_{bw} = 8$ frames.

| Parameters | | | All | | | Thing | | | Stuff | | | SSC | |
| $T_{fw}$ | $T_{bw}$ | $w$ | $PQ^{\dagger}\uparrow$ | $PQ\uparrow$ | SQ | RQ | PQ | SQ | RQ | PQ | SQ | RQ | IoU↑ | mIoU↑ |
|---|---|---|---|---|---|---|---|---|---|---|---|---|---|---|
| 8 | 0 | 2 | 7.79 | 0.75 | 36.62 | 1.32 | 1.41 | 46.97 | 2.42 | 0.42 | 31.44 | 0.77 | 13.48 | 8.82 |
| 16 | 0 | 2 | 9.67 | 2.91 | 34.13 | 5.07 | 2.46 | 46.79 | 4.32 | 3.13 | 27.80 | 5.44 | 18.98 | 10.97 |
| 32 | 0 | 2 | 10.48 | 4.11 | 34.86 | 6.76 | 2.73 | 47.76 | 4.74 | 4.80 | 28.41 | 7.77 | 22.02 | 12.08 |
| 48 | 0 | 2 | 10.49 | 4.15 | 34.86 | 6.80 | 2.74 | 47.85 | 4.75 | 4.85 | 28.37 | 7.83 | 22.09 | 12.10 |
| 32 | 0 | 2 | 10.48 | 4.11 | 34.86 | 6.76 | 2.73 | 47.76 | 4.74 | 4.80 | 28.41 | 7.77 | 22.02 | 12.08 |
| 32 | 4 | 2 | 11.65 | 4.80 | 41.03 | 7.77 | 3.53 | 56.61 | 6.23 | 5.43 | 33.24 | 8.54 | 24.53 | 13.32 |
| 32 | 8 | 2 | 12.21 | 5.08 | 40.84 | 8.20 | 3.96 | 47.44 | 6.97 | 5.63 | 37.55 | 8.81 | 25.64 | 13.93 |
| 16 | 0 | 4 | 10.97 | 4.13 | 37.28 | 7.04 | 2.98 | 50.04 | 5.35 | 4.71 | 28.39 | 7.88 | 22.10 | 12.55 |
| 32 | 8 | 2 | 12.21 | 5.08 | 40.84 | 8.20 | 3.96 | 47.44 | 6.97 | 5.63 | 37.55 | 8.81 | 25.64 | 13.93 |
| 64 | 16 | 1 | 13.10 | 5.69 | 47.59 | 9.07 | 5.34 | 57.04 | 9.33 | 5.87 | 42.87 | 8.94 | 27.96 | 14.81 |

*Table 5.* **Coverage analysis.** Coverage of mask pseudo-labels (w/o CRF, *Label*) and binary occupancy (w/o 360° aggr., *Occ.*).

| | SemanticKITTI(val) | | | | SSCBench-KITTI360 (val) | | | |
| CRF | Label | 360° | Occ. | | CRF | Label | 360° | Occ. |
| | Coverage % | Aggr. | Coverage % | | | Coverage % | Aggr. | Coverage % |
|---|---|---|---|---|---|---|---|---|
| ✗ | 28.04 | ✗ | 37.36 | | ✗ | 27.38 | ✗ | 44.15 |
| ✓ | 70.13 | ✓ | 99.96 | | ✓ | 50.48 | ✓ | 67.63 |

the baselines leverage completion models trained on fully completed GT, our approach excels despite using pseudo-labels with only partial coverage. This highlights that zero-shot panoptic Lidar scene completion is challenging and not trivially addressed by prior art.

**Completion and amodal 3D detection.** Our method completes the full amodal extent of objects, highlighting its potential for zero-shot amodal perception tasks. We qualitatively assess **CAL** for amodal 3D object detection in Fig. 5 by fitting 3D bounding boxes to the recognized objects.

### 4.3. Pseudo-labeling engine analysis

**Temporal mask aggregation.** In Tab. 4, we evaluate pseudo-label quality w.r.t GT labels, using PSC metrics in the SO setting (ZS results are provided in Appx. C.2). We ablate the number of frames used for forward $T_{fw}$ and backward $T_{bw}$ propagation and stride $w$. We perform this analysis *before* CRF refinement, directly using the output from Fig. 2 ③. We notice that propagation is beneficial in both forward and backward directions. We observe an increasing improvement up to $T_{fw} = 32$ (10.48 $PQ^{\dagger}$)

and only marginal improvements using $T_{fw} = 48$ (10.49 $PQ^{\dagger}$). Increasing $T_{bw}$ also improves performance. We find that the the best combination is $T_{bw} = 8$ and $T_{fw} = 32$ (12.21 $PQ^{\dagger}$). Interestingly, backward propagation improves completion of partially visible instances near the reference camera. We observe no significant improvements between $w = \{1, 2\}$ and a degradation in performance when increasing to $w = 4$ (10.97 $PQ^{\dagger}$) due to the large temporal gap the video propagation model (Ravi et al., 2024) must bridge. While the best results are achieved with $T_{fw} = 64$, $T_{bw} = 16$, $w = 1$ (13.10 $PQ^{\dagger}$), we use the combination $T_{fw} = 32$ $T_{bw} = 8$, $w = 2$ in our experiments due to the improved runtime of pseudo-labeling.

**Refinement with CRF.** We study the pseudo-label quality with & w/o CRF refinement (Fig. 2 ⑤) and report results in Tab. 3. Due to the limited visibility within a camera frustum, our pseudo labels cover approximately 28% of the binary occupancy obtained with full 360° Lidar scans (Tab. 5). CRF refinement greatly improves pseudo-label quality on SemanticKITTI and SSCBench-KITTI360 datasets (Tab. 3). For instance, we notice an improvement from 12.78 $PQ^{\dagger}$ to 25.90 $PQ^{\dagger}$ in the SO setting on SemanticKITTI.

**Pseudo-label coverage.** In Tab. 5, we study the pseudo-label coverage (*Label Coverage %*) with and w/o CRF refinement and binary occupancy coverage (*Occ. Coverage %*) with and w/o full LiDAR scan aggregation. Pseudo-label coverage is lower without CRF due to limited visi-

*Table 6.* **CAL model ablations.** We analyze the contribution of **CAL**'s key design choices and components: training $B_o$ with partial coverage ($B_o^{pc}$) or with full coverage ($B_o^{fc}$), introducing $S$, and adding $B_s$. Introducing $S$ provides a significant improvement, likely due to its implicit semantic regularization. Training with full coverage ($B_o^{fc}$) and $B_s$ further improve performance.

| Training Components | | | | All | | | | Thing | | | Stuff | | | SSC |
|---|---|---|---|---|---|---|---|---|---|---|---|---|---|---|
| $B_o^{pc}$ | $B_o^{fc}$ | $S$ | $B_s$ | PQ$^\dagger\uparrow$ | PQ$\uparrow$ | SQ | RQ | PQ | SQ | RQ | PQ | SQ | RQ | mIoU$\uparrow$ |
| *Semantic oracle* | | | | | | | | | | | | | | |
| ✓ | ✗ | ✗ | ✗ | 4.81 | 0.01 | 8.42 | 0.03 | 0.00 | 7.10 | 0.00 | 0.02 | 9.38 | 0.04 | 4.84 |
| ✓ | ✗ | ✓ | ✗ | 16.08 | 5.25 | 41.83 | 8.67 | 3.46 | 51.62 | 5.60 | 6.54 | 34.72 | 10.90 | 20.40 |
| ✗ | ✓ | ✓ | ✗ | 17.01 | 6.02 | 44.69 | 9.71 | 3.14 | 51.08 | 5.11 | 8.12 | 40.05 | 13.05 | 21.08 |
| ✗ | ✓ | ✓ | ✓ | 17.12 | 6.27 | 43.40 | 10.06 | 3.48 | 44.39 | 5.65 | 8.30 | 42.67 | 13.27 | 20.71 |
| *CLIP Semantics* | | | | | | | | | | | | | | |
| ✓ | ✗ | ✗ | ✗ | 3.73 | 0.01 | 5.68 | 0.01 | 0.00 | 7.10 | 0.00 | 0.01 | 4.66 | 0.02 | 3.15 |
| ✓ | ✗ | ✓ | ✗ | 11.98 | 4.21 | 21.19 | 6.92 | 1.81 | 22.46 | 2.82 | 5.96 | 20.27 | 9.91 | 11.96 |
| ✗ | ✓ | ✓ | ✗ | 13.43 | 5.10 | 33.10 | 8.16 | 1.70 | 23.73 | 2.62 | 7.57 | 39.92 | 12.19 | 12.48 |
| ✗ | ✓ | ✓ | ✓ | 13.12 | 5.26 | 27.45 | 8.44 | 2.42 | 22.79 | 3.89 | 7.33 | 30.84 | 11.76 | 13.09 |

bility with the camera frustums (Sec. 3.1). CRF improves this coverage by $1.9\times$ on SSCBench-KITTI360 and $2.5\times$ on SemanticKITTI. Similarly, binary occupancy coverage benefits from full Lidar scan aggregation, improving coverage from $37.36\%$ to $99.96\%$ on SemanticKITTI and from $44.15\%$ to $67.63\%$ on SSCBench-KITTI360[2].

### 4.4. **CAL** model analysis

**Model ablations.** In Tab. 6, we analyze the effects of the key **CAL** design choices and components: training $B_o$ with partial coverage ($B_o^{pc}$), training $B_o$ with full coverage ($B_o^{fc}$), introducing the (pseudo) semantic head $S$, and the additional binary head $B_s$. Tab. 6 reports the SO and ZS results for **CAL** trained on the SemanticKITTI dataset; we observe similar results for SSCBench-KITTI360. Training with partial coverage pseudo-labels ($B_o^{pc}$) achieves the worst performance of 4.81 $PQ^\dagger$ due to the model overfitting on common and fully visible instances. Introducing $S$ improves the results to 16.08 $PQ^\dagger$ and 11.98 $PQ^\dagger$ in the SO and ZS settings, respectively. This highlights that CLIP feature prototypes are beneficial for learning class-agnostic shape priors. Introducing full coverage guidance ($B_o^{fc}$) and $B_s$ improves performance in most settings.

**CLIP prototypes.** We study **CAL** performance when varying the number of CLIP prototypes $C$ (Sec. 3.2) and report results on SemanticKITTI (ZS) in Tab. 7. We observe that the performance is not particularly sensitive to the number of clusters $C \in \{6, 18, 50, 100\}$. Specifying $C = 18$ clusters (close to the number of annotated semantic groups in common datasets) yields the highest overall $PQ$. The extreme case of $C = 1$ shows significant performance degradation, demonstrating the benefits of CLIP prototypes for learning semantic occupancy priors. Unsurprisingly, performance is negatively affected when trained with too many clusters ($C = 500$). In practice, the number of clusters does

---

[2]The discrepancy likely originated from undocumented point accumulation strategy, which we could not clarify with authors.

*Table 7.* **Number of CLIP prototypes.** We evaluate SSC/PSC performance on SemanticKITTI when varying the number of CLIP prototypes $C$. We observe similar performance with $C \in \{6, 18, 50, 100\}$, indicating general robustness to $C$. Extreme cases ($C = 1$ and $C = 500$) result in performance degradation.

| | All | | | | Thing | | | Stuff | | | SSC |
|---|---|---|---|---|---|---|---|---|---|---|---|---|
| $C$ | PQ$^\dagger\uparrow$ | PQ$\uparrow$ | SQ | RQ | PQ | SQ | RQ | PQ | SQ | RQ | mIoU$\uparrow$ |
| 1 | 3.73 | 0.01 | 5.68 | 0.01 | 0.00 | 7.10 | 0.00 | 0.01 | 4.66 | 0.02 | 3.15 |
| 6 | 12.22 | 4.74 | 24.36 | 7.61 | 2.16 | 22.92 | 3.30 | 6.61 | 26.41 | 10.75 | 11.07 |
| 18 | 13.12 | 5.26 | 27.45 | 8.44 | 2.42 | 22.79 | 3.89 | 7.33 | 30.84 | 11.76 | 13.09 |
| 50 | 12.62 | 4.69 | 27.98 | 7.60 | 1.83 | 23.93 | 2.82 | 6.77 | 30.93 | 11.07 | 12.57 |
| 100 | 13.16 | 4.81 | 32.82 | 7.90 | 1.82 | 29.07 | 2.83 | 6.99 | 35.55 | 11.59 | 12.55 |
| 500 | 4.76 | 1.59 | 5.65 | 2.82 | 0.00 | 0.00 | 0.00 | 2.75 | 9.75 | 4.88 | 4.45 |

not constrain the types of objects that can be completed and recognized. For example, if car and van are in the same pseudo-class, our instance decoder can distinguish them as individual instances, while their estimated CLIP token allows us to recognize them as instances of different classes. Finally, we note that our approach under-segments rare semantic classes due to k-means-based CLIP feature space quantization.

**Limitations.** While our method demonstrates strong performance under zero-shot conditions, several limitations remain. Pseudo-label accuracy depends on the robustness of video foundation models, which can suffer from tracking errors such as ID switches and recognition failures, especially under significant view changes or occlusions. These errors can lead to incomplete or noisy label coverage, particularly in poorly visible regions. Another limitation is the performance gap between our zero-shot approach and fully-supervised methods, driven by challenges in zero-shot recognition and lower pseudo-label coverage due to camera-LiDAR visibility constraints. To address these limitations, future work could explore improved temporal association using geometric cues, integration of improved foundation models, and strategies for scaling to larger unlabeled datasets to further close the gap with supervised baselines.

## 5. Conclusion

We propose the first method for *Zero-Shot Lidar Panoptic Scene Completion*, capable of completing objects in a sparse and incomplete *Lidar scan*. The key components of **CAL** are a pseudo-labeling engine that mines completed shape priors with queryable semantic features, and a zero-shot model trained on pseudo-labels. Our work takes a step towards learning shape priors from temporal context, laying the foundation for amodal perception. However, our pseudo-labeling engine remains computationally expensive, and despite CRF refinement, label coverage is limited. A potential solution is leveraging self-supervised LiDAR forecasting to improve label coverage in fully unobserved regions. We believe these are promising directions for future work.

## Acknowledgements

We thank Zan Gojcic and Dávid Rozenberszki for their valuable feedback and comments!

## Impact Statement

We present a method for performing zero-shot object completion in Lidar scans, with possible applications in robotics and autonomous driving. Our method benefits from existing large vision-language models and vision foundation models, and relies on their generalization ability to connect text queries to our model's zero-shot object-completion output. Our work is intended as an initial step towards achieving such abilities, and we acknowledge that in its current form our method shares the limitations of the foundation models we rely on, restricting its operational robustness in safety critical environments. As these are not specifically our method's limitations but instead limitations of the foundation models, such consequences are already discussed in detail in the respective technical reports of the foundation models we employ. We believe there are no particular societal consequences we must specifically highlight here.

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

# A. `CAL` Pseudo-Labeling Engine Details

In this section, we provide additional implementation details of our pseudo-labeling engine, and provide further insights into the pseudo-labeled data we generate for training purposes. An overview of the pseudo-label generation configuration can be found in Tab. 8 for both SemanticKITTI (Behley et al., 2019) and KITTI360 (Liao et al., 2021) datasets.

*Table 8.* Pseudo-labeling engine configuration with dataset-specific parameters for SemanticKITTI and KITTI360.

| Parameter | SemanticKITTI | KITTI360 |
|---|---|---|
| *Temporal Window for Video-Object Localization* | | |
| Number of forward frames ($T_{fw}$) | 32 | 32 |
| Number of backward frames ($T_{bw}$) | 8 | 8 |
| Stride ($w$) | 2 | 2 |
| *Temporal Window for Binary Occupancy Aggregation* | | |
| Number of forward frames | 72 | 72 |
| Number of backward frames | 1 | 36 |
| Stride | 1 | 1 |
| *CRF Settings* | | |
| Number of iterations | 5 | 5 |
| *Other Settings* | | |
| Dynamic object removal | ✗ | ✓ |
| *SAM Automatic Mask Generator Configuration* | | |
| Model | sam_vit_h_4b8939 | sam_vit_h_4b8939 |
| Points per side | 32 | 32 |
| Points per batch | 64 | 64 |
| Pred. IoU threshold | 0.84 | 0.88 |
| Stability score threshold | 0.86 | 0.90 |
| Stability score offset | 1.0 | 1.0 |
| Crop N layers | 1 | 0 |
| Box NMS threshold | 0.7 | 0.8 |
| Crop N layers downscale factor | 2 | 1 |
| Min. mask region area | 100 | 100 |
| Crop NMS threshold | 0.7 | 0.8 |
| *SAM2 Video-Object Localization Configuration* | | |
| Model | sam2_hiera_large | sam2_hiera_large |
| Points per side | 32 | 32 |
| Points per batch | 64 | 64 |
| Pred. IoU threshold | 0.5 | 0.8 |
| Stability score threshold | 0.5 | 0.9 |
| Stability score offset | 1.0 | 1.0 |
| Crop N layers | 1 | 1 |
| Box NMS threshold | 0.7 | 0.7 |
| Crop N layers downscale factor | 2 | 2 |
| Min. mask region area | 100 | 100 |
| Crop NMS threshold | 0.7 | 0.7 |
| *Voxelization and Alignment Settings* | | |
| Voxel size | 0.2 | 0.2 |
| Scene size (m) | (51.2, 51.2, 6.4) | (51.2, 51.2, 6.4) |
| Voxel grid origin | (0, -25.6, -2) | (0, -25.6, -2) |
| Grid size | (256, 256, 32) | (256, 256, 32) |
| Camera-to-Lidar alignment shift | (0.0, 0.0, 0.0) | (0.79, 0.3, -0.25) |

### A.1. CRF-based refinement module

In our pseudo-labeling engine (as depicted in Fig. 2 of the main paper), we first accumulate two types of information: binary occupancy, and completed instance masks. As depicted in Tab. 10, the binary occupancy (voxels that are occupied) information we obtain has much higher coverage compared to our pseudo-labeled mask coverage. As the mask-level information relies on whether we were able to successfully detect and track a mask across the RGB sequence, there are areas in the full scene for which we only have occupancy information from the Lidar scan, but no intance-level labels. For example, if the ego-vehicle drives by a parked car, the RGB camera only observes the visible parts of the car, hence only tracking those areas across the video. This often results in incomplete mask coverage of objects that are only observed from one side, such as parked cars. However, our occupancy signal is more complete as it is obtained from 360° Lidar scans. To benefit from this additional information and improve our partial pseudo-label coverage, we employ Dense Conditional Random Fields (DenseCRF) to propagate our mask-level pseudo-labels to unlabeled but occupied areas, in order to improve and refine our masks. For this purpose, we rely on the DenseCRF Python wrapper provided in *pydensecrf*, based on the work of Krähenbühl et al. (Krähenbühl & Koltun, 2011).

We perform CRF-guided refinement only in the occupied area by operating on the sparse voxel representation. This implies that CRF does not change the occupancy coverage, but only increases mask-level label coverage. We use a multi-class setting where we define each separate instance mask ID as a class label (e.g. mask 1 has class ID 1 and mask 2 has class ID 2 etc.) in a class-agnostic setting. The unary energy is obtained from class-probabilities which represent the initial confidence in each voxel belonging to a particular class. For voxels which already belong to a mask, we initialize unary-potentials based on the original instance mask IDs, meaning that we set a probability of 1.0 for the original class. For the occupied but unlabeled voxels, we initialize the probabilities as a uniform distribution over all classes (i.e., instance mask IDs). Pairwise potentials for DenseCRF are defined based on the voxel coordinates, using features formulated as the normalized voxel coordinates. This way, the features represent the spatial proximity of the voxels. These pairwise potentials encourage neighboring voxels to have similar labels, thereby refining the initial pseudo labels, and improving the level of completion. We then run DenseCRF inference for 5 steps to refine the labels iteratively. Finally, we ensure that none of the original voxel mask labels change after the CRF-guided refinement by ensuring that the voxels with initial labels preserve their original mask labels. Furthermore, the only update that the CRF-guidance performs is regarding the class-agnostic masks over the voxel grid, meaning that the CLIP features are not affected by this operation.

### A.2. Handling dynamic objects

Our video-object localization process segments points that correspond to both static *and* dynamic parts of the scene. However, since we need to correctly accumulate a dense point cloud representation for the objects in the scene, handling dynamic objects would require registering their partial observations and correctly mapping them to the canonical pose for accurate reconstruction. Existing methods that perform dynamic object synchronization and rectification (Li et al., 2024; Xia et al., 2023) rely on panoptic segmentation labels for identifying and registering such objects. However, (1) we assume no access to semantic labels, and (2) performing such registration is error-prone in our setting.

In the original SemanticKITTI completion benchmark (Behley et al., 2019), while performing point level aggregation, spatio-temporal tubes for dynamic objects are preserved, and such objects are neither registered to the canonical pose and synchronized per frame, nor removed. For consistency with the supervised baselines, we perform the same, where we preserve the moving-object tubes. However, for the KITTI-360 (Liao et al., 2021) dataset where the 3D object tracks are available as an asset in the dataset, we perform a dynamic object removal operation and only keep the static parts of the scene during Lidar point cloud accumulation. We would like to highlight that while such object tracks are commonly available in most autonomous driving datasets, it is also possible to instead use automatic 3D object detection approaches for this purpose. For this dynamic object removal operation, after backprojecting the predicted 2D instance masks onto the Lidar point cloud, we first fit 3D z-axis aligned bounding boxes to each 3D object point cloud. If the overlap between the 3D bounding boxes of dynamic objects (from the 3D object tracks available as an asset) and the predicted (fitted) object boxes exceeds a certain limit, we classify this predicted object as *dynamic*. Any object mask ID associated with a dynamic object in any frame within the temporal window $[t - T_{bw}, t + T_{fw}]$ is discarded, and not used for final aggregation of pseudo-labeled instance masks. To note once again, we perform this process only for KITTI360, and we keep the moving object tubes for our experiments in SemanticKITTI, in alignment with the original benchmark from (Behley et al., 2019). Our results show that our method **CAL** learns to complete both static and dynamic objects despite the fact that only the static parts get correctly aligned (registered) for completion in our pseudo-labeled training data for both datasets.

### A.3. Lifting and refinement

Following Osep et al. (2024), we perform a refinement of the 3D masks $m_{t,k}^{3D}$ based on 3D segments obtained via DBSCAN clustering on the points from the Lidar scan $L_t$, which are inherently less noisy in the 3D space. More specifically, for each Lidar scan, we perform DBSCAN at varying density thresholds $\epsilon \in \{(1.2488, 0.8136, 0.6952, 0.594, 0.4353, 0.3221)\}$ to compensate for varying point density in Lidar scans. This operation is performed after ground plane fitting following Osep et al. (2024). After obtaining clusters (segments) with each of these thresholds, we match each predicted 3D mask $m_{t,k}^{3D}$ with the best-matching segment (matching in terms of IoU). If the overlap between the predicted mask and the matched segment is above a certain threshold (0.5), we replace each predicted mask $m_{t,k}^{3D}$ with the matched segment. Differently from (Osep et al., 2024), we perform this operation for each scan in our temporal window $[t - T_{bw}, t + T_{fw}]$.

### A.4. Voxelization details

Following the voxelization process from SemanticKITTI (Behley et al., 2019), which was also followed in SSCBench-KITTI360 (Li et al., 2024), we crop the volume in front of the camera using a minimum extent of $[0, -25.6, -2]$ and maximum extent of $[51.2, 25.6, 4.4]$ in meters. Each aggregated point cloud is first cropped within these bounds, and then centralized using the predefined voxel-origin, by subtracting $(0, -25.6, -2)$ from the aggregated point coordinates. Next, we use a voxel size of 0.2 following prior work to voxelize the point coordinates obtained in the previous step. For SSCBench-KITTI360, we observe that the camera placement procedure is different than the procedure in SemanticKITTI, as the ground truth voxel grids had undergone a Lidar-to-Camera alignment transformation according to the discussions available in the official repository of SSCBench-KITTI360. However, we were unable to confirm the exact transformation process after reaching out, so we empirically determined a transformation vector of $[0.79, 0.3, -0.25]$ with our best efforts to obtain an alignment vector to apply to our pseudo-labels before voxalization, in order to provide correct GT alignment for evaluation, and to have a fair comparison between our method and the supervised baselines.

### A.5. CLIP features

To obtain CLIP features associated with each object instance, we employ MaskCLIP (Zhou et al., 2022) on each masklet across the video sequence, and average-pool normalized CLIP features obtained at each timestamp, per object. The output from this operation is a separate CLIP feature vector of dimension 768, for each object in the scene.

## B. `CAL` Model Details

### B.1. Architecture Details

**Overview.** Our model architecture is built upon a sparse-generative 3D U-Net, largely following the implementation in (Cao et al., 2024), which consists of a feature backbone, an encoder followed by a dense 3D convolutional network, and a multi-scale generative decoder consisting of 3 decoder blocks for each of the 3 scales of resolution. The features from the generative decoder are passed to a transformer decoder which aims to produce a set of binary instance masks over the predicted occupied voxels, along with an associated CLIP feature for each mask.

**Encoder.** The input to our model is a Lidar point cloud that is cropped within the bounding volume of the voxel-grid, which is represented as an unstructured set of coordinates. This cropping operation follows the problem formulation introduced for SSC and PSC benchmarks (Cao et al., 2024). This input point cloud is given as input to MLP and voxelization layers based on Cylinder3D (Zhu et al., 2021) as in PaSCo (Cao et al., 2024). The voxelized features are then passed through a sparse convolutional encoder that is composed of 4 encoder blocks which gradually upsample the features, resulting in $f_{enc}^{1:1}, f_{enc}^{1:2}, f_{enc}^{1:4}, f_{enc}^{1:8}$. First encoder block consists of 3 residual blocks, and each of the remaining 3 encoder blocks consists of a sparse convolutional layer (intended for feature upsampling) and 3 residual blocks. The final output from the encoder is 1:8 resolution features $f_{enc}^{1:8}$. Since the original input is sparse, and we would like to perform scene-scale completion in a dense voxel grid, we employ a dense 3D convolutional network following PaSCO. This dense 3D convolutional network takes $f_{enc}^{1:8}$ from the encoder, and outputs the features $f_{d3D}$.

**Multi-scale generative decoder.** The decoder in our sparse-generative 3D U-Net consists of 3 decoder blocks $D^{1:L}$, for each scale $L \in [4, 2, 1]$. Each decoder block $D^{1:L}$ is a structured generative block designed for sparse tensor operations. It begins with an upsampling operation using a transposed convolution to increase spatial resolution, followed by batch

normalization and a LeakyReLU activation layer. The resizing step adjusts feature dimensions via a point-wise convolution. The output features from the decoder block $D^{1:L}$ is denoted as $f_{dec}^{1:L}$.

**Completion heads.** The output features from each decoder block at each scale is passed through a set of completion heads: binary occupancy prediction heads ($B_o^{1:L}$ and $B_s^{1:L}$, as well as $S^{1:L}$) for prototype prediction formulated as a classification task. Please note that in our setting, we only use CLIP prototypes as a proxy for semantic classes, but we never have access to the ground truth semantic classes at any capacity. To provide further detail, we have two types of binary occupancy prediction heads: $B_o^{1:L}$ and $B_s^{1:L}$. The binary occupancy head $B_o^{1:L}$ is a sparse convolutional layer which directly takes decoder features $f_{dec}^{1:L}$, and predicts per-voxel binary occupancy. $B_s^{1:L}$ on the other hand, takes the semantic voxel logits from the semantic completion head $S^{1:L}$, i.e. $S^{1:L}(f_{dec}^{1:L})$ as input. The reason why we have the additional $B_s^{1:L}$ head is the following: since our binary occupancy labels are more complete, whereas the pseudo-labeled area with instance masks and CLIP features have a smaller amount of coverage compared to the full occupied area, we are supervising binary occupancy and semantic head with two separate completion signals with differing volume coverage. Therefore, by adding an additional binary occupancy head to the output of the prototype-classification head, we aim to ensure that the classification head output will not diverge from the binary occupancy head output, which could potentially result in mismatched regularization. This way, we intend to encourage the network to complete the scene even in areas for which we do not have pseudo-labels (masks and associated CLIP features).

**Pruning.** Similar to Cao et al. (2024), we also perform pruning of voxels based on the logits predicted by the prototype-classification head to preserve sparsity, and continue refining per-voxel predictions only for voxels that were predicted to be occupied in the coarser resolution. Further details regarding the pruning could be found in (Cao et al., 2024). The decoder block output from a coarser scale is processed through the pruning layer, and the output is then passed to the next decoder block as input. This process is repeated until the final decoder block with the highest resolution.

**Transformer predictor.** We adopt the mask-centric transformer architecture from (Cao et al., 2024), which is based on the multi-scale decoder layer from Mask2Former (Cheng et al., 2022). This transformer model aims to perform panoptic scene completion based on the multi-scale features $f_{dec}^{1:L}$ extracted in the generative decoder backbone. We perform the operations only on the voxels that are predicted to be occupied in the generative decoder. The transformer decoder consists of 3 layers. In addition to the original mask prediction heads from (Cao et al., 2024) our transformer decoder also includes a CLIP distillation head which aims to predict a CLIP token for each query. This MLP block consists of layers with dimensions $[384, 512, 1024, 768, 768]$, where the final dimension, 768, is the dimensionality of the CLIP embedding space. In essence, for each query, our method regresses a CLIP feature vector. Importantly, for mask prediction, our method outputs a soft mask over the voxels: e.g., per-voxel probability over the list of sparse voxels (which are the voxels predicted to be occupied) for a voxel to be included in the instance mask.

## B.2. Implementation Details

**Processing the predictions.** A voxel probability threshold parameter ($\tau_{vox}$ is applied to obtain the final binary instance mask. The *objectness* of the predicted instance mask is computed as the average per-voxel probability of the predicted instance mask. We only keep the instance masks with an objectness score that is at least $\tau_{obj}$. Lastly, we suppress overlapping masks, if the pairwise mask overlap is above a specified mask overlap threshold, $\tau_{ovr}$. The mask overlap threshold, objectness threshold and voxel probability threshold parameters are used both during training and test to refine the output. Since with our pseudo-labeled dataset with occasional incomplete observations, we expect to have in general lower confidence in the predicted masks and completed voxels. We empirically find that with the SemanticKITTI dataset which consists of a smaller number of training samples, we benefit from setting lower confidence thresholds during inference. Therefore, while performing inference for SemanticKITTI, we use an overlap threshold of $\tau_{ovr} = 0.1$, an objectness threshold of $\tau_{obj} = 0.1$, and voxel probability threshold of $\tau_{vox} = 0.1$. For the KITTI-360 dataset which has a higher number of samples, we use an overlap threshold of $\tau_{ovr} = 0.4$, and objectness threshold of $\tau_{obj} = 0.5$ and voxel probability threshold of $\tau_{vox} = 0.3$.

**CLIP prototypes.** To obtain the fixed prototype centers as described also in the main paper, we perform a clustering of the CLIP features associated with each instance mask we have in our pseudo-labeled dataset. For SemanticKITTI, we have 118333 individual CLIP feature instances in our pseudo-labeled dataset, and for for KITTI-360 we have 301332 CLIP feature instances. For clustering these features, we apply $k$-means, defining the number of groups we expect to create. We experiment with different numbers of clusters. The setting $k=1$ refers to treating all masks with the same prototype class, which in essence reduces to a trivial classification problem, reducing to overall task of the completion heads to binary

occupancy prediction. We also use the setting *k=6* as a proxy for supergroups of expected number of classes, following Osep et al. (2024), and *k=18* which represents the general number of commonly annotated number of classes for urban scene datasets for autonomous driving such as SemanticKITTI and KITTI-360. We also experiment with higher numbers of clusters to assess our method's sensitivity to the number of clusters, by setting $k \in 50, 100, 500$. For all settings, the prototypes are fixed at the beginning, and are not updated during training. At inference time, we completely discard our model's prototype class predictions, and only use the predicted instance masks over the voxel grid as well as the predicted CLIP features.

### B.3. `CAL` Training Details

**Overall Loss Definition.** Following the notation in the main paper, our total training objective combines four main terms: (i) binary occupancy completion loss $\mathcal{L}_{\text{occ}}$, (ii) class-agnostic instance mask prediction loss $\mathcal{L}_{\text{mask}}$, (iii) CLIP feature distillation loss $\mathcal{L}_{\text{CLIP}}$, and (iv) per-voxel prototype classification loss $\mathcal{L}_{\text{prot}}$.

Total loss is formulated as

$$\mathcal{L}_{\text{total}} = \lambda_{\text{occ}} \mathcal{L}_{\text{occ}} + \lambda_{\text{prot}} \mathcal{L}_{\text{prot}} + \lambda_{\text{mask}} \mathcal{L}_{\text{mask}} + \lambda_{\text{CLIP}} \mathcal{L}_{\text{CLIP}} + \mathcal{L}_{\text{aux}}, \tag{1}$$

where each $\lambda$ is a scalar weight, $\lambda_{\text{compl}} = 1.0$, $\lambda_{\text{mask}} = 40.0$, $\lambda_{\text{CLIP}} = 1.0$, $\lambda_{\text{prot}} = 1.0.$. $\mathcal{L}_{\text{aux}}$ is an auxiliary loss term following Cao et al. (2024). Below, we provide a detailed overview of the loss terms.

**Completion Losses, $\mathcal{L}_{\text{occ}}$ and $\mathcal{L}_{\text{prot}}$.** As described in the main paper, we supervise both geometric occupancy and coarse semantic structure (via CLIP-based prototypes) at multiple scales. Specifically, we define **binary occupancy loss** ($\mathcal{L}_{\text{occ}}$) as the term which supervises whether each voxel is occupied or empty via binary cross entropy. Next, we define $\mathcal{L}_{\text{prot}}$, which is the **per-voxel prototype classification loss** which classifies each voxel into one of $C$ prototype categories derived from CLIP features. This combines cross-entropy and Lovász losses. While computing this loss, we explicitly ignore voxels that do not have a corresponding pseudo-label associated with them, as the pseudo-labeling system often has a limited coverage of the full voxel grid volume. In practice, $\mathcal{L}_{\text{occ}}$ and $\mathcal{L}_{\text{prot}}$ terms are computed and averaged across 3 scales (from coarse to fine, $L \in \{4, 2, 1\}$) based on the outputs from the generative decoder. These multi-scale loss terms are averaged across 3 scales, i.e.

$$\mathcal{L}_{\text{prot}} = \frac{1}{|\{1, 2, 4\}|} \sum_{L \in \{1,2,4\}} \mathcal{L}_{\text{prot}}^{1:L} \quad \text{and} \quad \mathcal{L}_{\text{occ}} = \frac{1}{|\{1, 2, 4\}|} \sum_{L \in \{1,2,4\}} \mathcal{L}_{\text{occ}}^{1:L}.$$

More specifically,

$$\mathcal{L}_{\text{occ}}^{1:L} = \mathcal{L}_{\text{occ,B}_o}^{1:L} + \mathcal{L}_{\text{occ,B}_s}^{1:L}$$

with individual binary occupancy loss terms for $B_o$ and $B_s$, where $\mathcal{L}_{\text{occ,B}_o}^{1:L}$ and $\mathcal{L}_{\text{occ,B}_s}^{1:L}$ are defined as a weighted binary cross entropy (wBCE) loss terms with respect to the pseudo-label binary occupancy at level L, $O_{label}^{1:L}$, and predicted occupancy outputs from $B_o$ and $B_s$, defined as $O_{B_o}^{1:L}$ and $O_{B_s}^{1:L}$ respectively:

$$\mathcal{L}_{\text{occ,B}_o}^{1:L} = wBCE(O_{label}^{1:L}, O_{B_o}^{1:L}) \quad \text{and} \quad \mathcal{L}_{\text{occ,B}_s}^{1:L} = wBCE(O_{label}^{1:L}, O_{B_s}^{1:L}).$$

In our implementation, we use a class weight of 20 for the occupied voxels in the weighted binary cross-entropy computation.

**Instance Mask-Matching Loss $\mathcal{L}_{\text{mask}}$.** Our transformer decoder predicts class-agnostic instance masks over the completed 3D volume. To compute $\mathcal{L}_{\text{mask}}$, we perform a one-to-one Hungarian matching between predicted masks and pseudo-labeled instance masks. Unlabeled voxels are ignored both during Hungarian matching as well as mask loss computation to avoid penalizing predictions outside the limited pseudo-labeled regions. Following previous work on 3D instance-segmentation, we define:

$$\mathcal{L}_{\text{mask}} = \lambda_{\text{CE}} \mathcal{L}_{\text{CE}} + \lambda_{\text{Dice}} \mathcal{L}_{\text{Dice}},$$

where $\mathcal{L}_{\text{CE}}$ and $\mathcal{L}_{\text{Dice}}$ are per-voxel binary cross-entropy and Dice terms, respectively, and $\lambda_{\text{CE}}$, $\lambda_{\text{Dice}}$ are scaling factors. We set $\lambda_{\text{CE}} = 2.0$, $\lambda_{\text{Dice}} = 1.0$ in our experiments and apply an overall factor $\lambda_{\text{mask}}$ when adding to (1).

**CLIP Distillation Loss $\mathcal{L}_{\text{CLIP}}$.** Simultaneously, our transformer regresses CLIP features per predicted instance to align with the pseudo-labeled embeddings, as also described in the main text. We use a cosine embedding loss between predicted

CLIP embeddings, and the pseudo-label GT embedding associated with the matched instance mask:

$$\mathcal{L}_{\text{CLIP}} = \text{CosineEmbeddingLoss(pred\_embeds, gt\_embeds)}.$$

This ensures that predicted instance-level embeddings preserve the semantic structure learned by CLIP. We weight $\mathcal{L}_{\text{CLIP}}$ by $\lambda_{\text{CLIP}}$ in (1).

**Further Implementation Details and Model Overview.** Our model consists of 118M trainable parameters in total, with 8.2M parameters allocated to the transformer predictor and 59.3K parameters for the CylinderFeat backbone. We train the model for 50 epochs on 8 NVIDIA A100 GPUs, using a batch size of 8 with 1 item per GPU, and a learning rate of 0.0001. As also described earlier in A.4, we follow (Behley et al., 2019) and define a voxel grid volume extending $51.2\,m$ forward, $25.6\,m$ to the sides, and $6.4\,m$ in height, with a voxel size of $0.2\,m$. We adopt this setting for both model training and inference. During training, we treat all pseudo-labeled instances equally, making no distinction between *stuff* and *things*—our approach is entirely zero-shot and class-agnostic. All unlabeled voxels are ignored during loss computation for $\mathcal{L}_{\text{mask}}$ and $\mathcal{L}_{\text{prot}}$, ensuring no penalty for predictions outside the spatial coverage of pseudo-labeled instances.

## C. Additional Evaluation Details and Analysis

### C.1. Dataset and Implementation Details

**Datasets and benchmarks.** We follow prior work (Cao et al., 2024) and evaluate **CAL** on two datasets that provide semantic *and* instance-level labels for PSC: SSCBench-KITTI360 (Li et al., 2024; Liao et al., 2021) and SemanticKITTI (Behley et al., 2019; Geiger et al., 2012; 2013) whose instance-level labels are provided in (Cao et al., 2024). These datasets provide per-voxel semantic labels (SemanticKITTI: 20 classes, 8 are *thing*; SSCBench-KITTI360: 19 classes, 6 are *thing*) that we only use during evaluation. Data was recorded using a 64-beam Velodyne Lidar sensor at $10\,Hz$, accompanied by a front RGB stereo camera (camera placement differs across the two datasets, that were recorded in different periods). We only utilize the left front RGB camera for pseudo-labeling, but we only use the Lidar scans at inference time, not requiring any camera data. During temporal accumulation of pseudo-labels (Fig. 2, ③), we identify and remove dynamic objects in KITTI-360 using 3D object tracks. For SemanticKITTI, we follow the original benchmark (Behley et al., 2019) and keep spatio-temporal moving object tubes for consistency with supervised baseline methods.

**Data statistics.** To have a fair comparison between our method and the supervised panoptic completion methods reported by Cao et al. (2024), we generate pseudo-labels for the same Lidar scans that were used for training these methods. More specifically, for SemanticKITTI, we use one scan in every 5 scans, resulting in a total of 4649 pseudo-label samples. For KITTI-360, we use the Lidar scan IDs used in SSCBench-KITTI360, resulting in 8487 training scans, 1780 validation scans, and 2165 test scans as pseudo-labels. While we limited our training data scale for fair comparison, our pseudo-labeling engine can label an arbitrary number of samples if desired, enabling easy scaling of data size.

**Text prompt tuning.** For our PSC and SSC experiments, we follow the prompt-tuning strategy from previous work (Osep et al., 2024), and we define per-class text prompts for SemanticKITTI and KITTI-360 by associating each semantic class with multiple potential queries describing the class, resulting in a more informative set of prompts. For instance, the query "car" can be addressed by "car, jeep, SUV, van" or the query "bicycle" can be addressed by "bicycle, bike". Full list of such queries can be found in "Table E.1: Dataset vocabulary text prompts" of SAL (Osep et al., 2024). We also employ the additional text prompt augmentation strategy from Osep et al. (2024) by including different variants of sentence structures describing the semantic class in an open-vocabulary setting. Overall similarity between a predicted CLIP feature and a semantic class is obtained by averaging the per-prompt similarity obtained for all prompts for this class.

### C.2. Additional Ablations for the Pseudo-Labeling Engine

In this section, we provide additional ablation studies and further analysis on our pseudo-labeling engine.

**Parameter analysis.** First, in Tab. 9, we present an ablation on the key hyperparameters governing the pseudo-label aggregation process. In the main paper, we provided the SO setting for this experiment to decouple tracking and completion performance from the CLIP feature quality. To complement this, here we share the ZS counterpart of this experiment. Particularly, we ablate the affect of the number of frames for mask propagation $T_{fw}$ (forward) and $T_{bw}$ (backward), as well as the stride, $w$. This table presents the performance metrics for various combinations of these parameters $(T_{fw}, T_{bw}, w)$

evaluated on the KITTI-360 validation set using PSC and SSC parameters $PQ^\dagger$, PQ, SQ, RQ, IoU, mIoU. We additionally report the relative number of time units required to obtain pseudo-labels for each setting, as well as the average pseudo-label coverage achieved with respect to the GT labels. To ensure consistency across different parameter settings, we perform these measurements *before* performing CRF.

*Table 9.* **Pseudo-labeling engine ablations using CLIP semantics.** This table presents an analysis on the key parameters of the pseudo-label aggregation process: the number of frames for tracking $T_{fw}$ and $T_{bw}$, as well as the stride, $w$.

| Parameters | | | All | | | | Thing | | | Stuff | | | SSC | | Time | Label |
|---|---|---|---|---|---|---|---|---|---|---|---|---|---|---|---|---|
| $T_{fw}$ | $T_{bw}$ | $w$ | $PQ^\dagger\uparrow$ | PQ↑ | SQ | RQ | PQ | SQ | RQ | PQ | SQ | RQ | IoU↑ | mIoU↑ | $t_k\downarrow$ | Cov.↑ |
| 8 | 0 | 2 | 6.30 | 0.35 | 21.94 | 0.63 | 0.57 | 37.90 | 1.01 | 0.24 | 13.96 | 0.43 | 13.47 | 5.54 | 1 | 14.50 |
| 16 | 0 | 2 | 7.91 | 2.17 | 24.99 | 3.78 | 1.26 | 46.71 | 2.26 | 2.63 | 14.12 | 4.55 | 18.97 | 7.28 | 2 | 20.32 |
| 32 | 0 | 2 | 8.61 | 3.25 | 25.70 | 5.33 | 1.34 | 47.97 | 2.37 | 4.21 | 14.56 | 6.81 | 22.00 | 8.23 | 4 | 23.31 |
| 48 | 0 | 2 | 8.70 | 3.34 | 25.75 | 5.45 | 1.34 | 48.06 | 2.37 | 4.34 | 14.59 | 6.99 | 22.07 | 8.34 | 6 | 23.32 |
| 32 | 0 | 2 | 8.61 | 3.25 | 25.70 | 5.33 | 1.34 | 47.97 | 2.37 | 4.21 | 14.56 | 6.81 | 22.00 | 8.23 | 4 | 23.31 |
| 32 | 4 | 2 | 9.32 | 3.71 | 28.85 | 5.95 | 1.82 | 56.40 | 3.25 | 4.66 | 15.08 | 7.30 | 24.51 | 8.96 | 4.5 | 26.13 |
| 32 | 8 | 2 | 10.04 | 3.96 | 25.80 | 6.32 | 2.25 | 47.15 | 4.01 | 4.82 | 15.12 | 7.48 | 25.62 | 9.32 | 5 | 27.38 |
| 16 | 4 | 4 | 8.73 | 3.17 | 27.68 | 5.37 | 1.45 | 54.73 | 2.67 | 4.03 | 14.15 | 6.72 | 22.09 | 8.37 | 2.5 | 23.47 |
| 32 | 8 | 2 | 10.04 | 3.96 | 25.80 | 6.32 | 2.25 | 47.15 | 4.01 | 4.82 | 15.12 | 7.48 | 25.62 | 9.32 | 5 | 27.38 |
| 64 | 16 | 1 | 10.63 | 4.41 | 28.98 | 6.93 | 3.11 | 57.07 | 5.48 | 5.06 | 14.93 | 7.65 | 27.94 | 9.87 | 10 | 29.98 |

**How well is our pseudo-label quality in the high-coverage area?** Our pseudo-labeled dataset covers only a limited portion of the full grid compared to the ground truth (70.14% for SemanticKITTI and 50.48% for KITTI-360) due to occasional failures in SAM2 mask detection or tracking across the video sequence, leading to missing label areas in the voxel grid. The panoptic scene completion evaluation using pseudo-labels penalizes these missing areas when performed over the full voxel grid (*full-grid eval*). To better assess the CLIP feature and instance-level completion performance of our pseudo-labels in areas for which we do have labels, we also evaluate performance restricted to areas with pseudo-labels by masking out excluded voxels (*masked-voxel eval*) in Tab. 10. The difference between full-grid and masked-voxel evaluations highlights that where completion signals exist, CLIP features and instance masks are generally informative and well-separated. However, the observed gap underscores the limited label coverage of pseudo-labels, which could be improved through enhancements in pseudo-label construction or training augmentations in future work.

*Table 10.* **Pseudo-label evaluation restricted to the areas in the voxel grid for which we have pseudo-labels.** Analysis of the accuracy of pseudo-labels on the SemanticKITTI (Behley et al., 2019) validation set. The *full-grid eval* setting refers to evaluating our pseudo-labels using the usual PSC evaluation with respect to the GT. The *masked-voxel eval* setting refers to excluding the voxels for which we don't have any pseudo-labels during PSC evaluation.

| | Semantic KITTI (Behley et al., 2019) (val set) | | | | | | | | | | |
|---|---|---|---|---|---|---|---|---|---|---|---|
| | | All | | | | Thing | | | Stuff | | |
| | $PQ^\dagger\uparrow$ | PQ↑ | SQ | RQ | PQ | SQ | RQ | PQ | SQ | RQ | mIoU↑ |
| *Pseudo-labels (full-grid eval)* | | | | | | | | | | | |
| **CAL** (Sem. oracle) | 25.90 | 17.71 | 64.06 | 26.55 | 16.93 | 67.91 | 23.75 | 18.28 | 61.25 | 28.58 | 33.10 |
| **CAL** (CLIP Semantics) | 16.98 | 10.67 | 54.02 | 16.19 | 7.30 | 67.19 | 10.21 | 13.12 | 44.43 | 20.54 | 20.62 |
| *Pseudo-label (masked-voxel eval)* | | | | | | | | | | | |
| **CAL** (Sem. oracle) | 33.20 | 26.09 | 68.05 | 36.62 | 20.57 | 69.39 | 28.21 | 30.10 | 67.07 | 42.74 | 42.37 |
| **CAL** (CLIP sem.) | 20.89 | 15.65 | 62.36 | 22.42 | 8.34 | 68.53 | 11.48 | 20.97 | 57.87 | 30.37 | 25.47 |

**Discussion on the possibility of pseudo-labeling efficiency.** Video-object localization over long video sequences is costly, particularly due to the costs associated with SAM2 (Ravi et al., 2024) mask propagation, which often results in a single pseudo-label pair to be generated in the order of a few minutes. Since this process can be costly, we also investigated whether it is possible to make the overall process more efficient by reducing the number of frames that one needs to process and aggregate for a single pseudo-labeled training sample. To do so, we experimented with reducing the number of required frames in conjunction with CRF-refinement, as in our earlier experiments we observed that CRF plays a significant role in improving the completion of scene objects. Therefore, we designed an experiment where, instead of using a long tracking horizon ($T_{fw} = 32, T_{bw} = 8$), we used a short tracking horizon but also performed CRF refinement. Tab. 11 presents our findings from this experiment on SemanticKITTI, using the SO evaluation setting. In the first block of Tab. 11, we propagate masks from reference frame for $T_{fw} = 8$ frames (FW), and compare to the second block with our original setting($T_{fw} = 32, T_{bw} = 8$), *after* CRF-based mask propagation. Here we see that while using a long-tracking horizon improves completion performance, it is also possible to instead use a shorter horizon in conjunction with CRF to reduce

video-object localization prcessing time. We believe that this observation might be relevant for future scaling efforts by saving processing time, and enabling the more sparse labeling of sequences for future data-scaling endeavors.

*Table 11.* Impact of the number of frames on completion quality with CRF on Semantic KITTI (Behley et al., 2019) (val). This study demonstrates that CRF significantly enhances completion quality, allowing the use of fewer frames in the pseudo-labeling pipeline which is computationally expensive due to the costly mask propagation step. The first block evaluates mask tracking over 8 frames, while the second block uses our original setting with 32 frames forward and 8 frames backward tracking.

| | | All | | | Thing | | | Stuff | | | | |
|---|---|---|---|---|---|---|---|---|---|---|---|---|
| | **PQ**$^\dagger$↑ | PQ↑ | SQ | RQ | **PQ** | SQ | RQ | **PQ** | SQ | RQ | mIoU | Label Cov.↑ |
| $T_{fw} = 8$, $T_{bw} = 0$, *with CRF* | | | | | | | | | | | | |
| **CAL** (SO) | 22.06 | 11.41 | 62.01 | 17.79 | 14.16 | 67.90 | 20.25 | 9.40 | 57.73 | 16.00 | 27.00 | 52.36 |
| **CAL** (CS) | 13.53 | 5.65 | 50.30 | 8.92 | 5.53 | 62.06 | 7.68 | 5.74 | 41.72 | 9.82 | 15.46 | 52.36 |
| $T_{fw} = 32$, $T_{bw} = 8$, *with CRF* | | | | | | | | | | | | |
| **CAL** (SO) | 25.90 | 17.71 | 64.06 | 26.55 | 16.93 | 67.91 | 23.75 | 18.28 | 61.25 | 28.58 | 33.10 | 70.13 |
| **CAL** (CS) | 16.98 | 10.67 | 54.02 | 16.19 | 7.30 | 67.19 | 10.21 | 13.12 | 44.43 | 20.54 | 20.62 | 70.13 |

## C.3. Additional Ablations and Analysis for the Model Performance

**Training data ablations for model training.** In the main paper, we discussed the improvements that CRF-refinement brings to pseudo-label quality, demonstrated in Tab. 3. In this section, we also discuss the effect of training with pseudo-labels with and without CRF-refinement on the model performance for completeness. The findings from this experiment are presented in Tab. 12 and Tab. 13. We observe that training with pseudo-labels that use CRF-refinement (resulting in higher label coverage) clearly improves PQ$^\dagger$ as well as PQ$^{thing}$, SQ$^{thing}$ and RQ$^{thing}$ results. However, we see that *Stuff* categories are negatively impacted by this variant for the SSCBench-KITTI360 dataset, which we suspect is due to CRF being relatively less effective in successfully propagating labels in KITTI360 dataset, where our binary occupancy coverage is only around 70% compared to the setting in SemanticKITTI where we reach almost 100% binary occupancy coverage. This is important, because CRF can only propagate labels towards unlabeled but occupied areas. Overall, it is evident that CRF refinement brings significant benefits to zero-shot PSC performance.

*Table 12.* **Model ablations for data on SemanticKITTI.** We train the **CAL** model using two different sets of data: pseudo-labels w/o CRF refinement, and pseudo-labels with CRF refinement. We report PSC metrics for both variants individually. We observe that training **CAL** with data that has higher label coverage, (pseudo labels w. CRF) clearly improves overall performance.

| Training data for **CAL** | | All | | | | Thing | | | Stuff | | | SSC |
|---|---|---|---|---|---|---|---|---|---|---|---|---|
| Setting | PQ$^\dagger$↑ | PQ↑ | SQ | RQ | PQ | SQ | RQ | PQ | SQ | RQ | mIoU↑ |
| *Pseudo Labels w/o CRF* | | | | | | | | | | | |
| Semantic oracle | 12.25 | 3.60 | 43.86 | 5.98 | 0.77 | 41.06 | 1.40 | 5.65 | 45.89 | 9.31 | 15.37 |
| CLIP semantics | 9.65 | 3.27 | 26.27 | 5.40 | 0.32 | 20.47 | 0.57 | 5.43 | 30.48 | 8.90 | 9.40 |
| *Pseudo Labels w. CRF* | | | | | | | | | | | |
| Semantic oracle | 17.12 | 6.27 | 43.40 | 10.06 | 3.48 | 44.39 | 5.65 | 8.30 | 42.67 | 13.27 | 20.71 |
| CLIP semantics | 13.12 | 5.26 | 27.45 | 8.44 | 2.42 | 22.79 | 3.89 | 7.33 | 30.84 | 11.76 | 13.09 |

**Pseudo-labels *vs*. model predictions.** In this section, we analyze the gap between our pseudo-labels and model predictions (both evaluated on the SemanticKITTI validation set; semantic classes are assigned based on CLIP-feature prompting). We note that our model is only given a single Lidar point cloud input and predicts occupancy, segmentation masks, and corresponding CLIP features used for prompting. On the other hand, pseudo-labels are obtained by (i) segmenting objects in images/video, (ii) averaging CLIP features, extracting from images, and (iii) accumulating Lidar measurements across a temporal window. Pseudo-labels, therefore, derive occupancy estimates from geometry observed from multiple views, and semantic features from visual information, while the model must rely on learned occupancy priors and semantic features distilled from the image- to the Lidar domain. In the semantic oracle setting, our model obtains 17.12 $PQ^\dagger$ (66.10 % of pseudo-labels), and 13.12 (77.27 % of pseudo-labels), suggesting that currently, the bottleneck is instance-level completion. Finally, we note that this gap is not surprising and largely stems from the inherent task difficulty- state-of-the-art PSC model (Cao et al., 2024) obtains 19.53 $PQ^\dagger$ (26.29 when ensembling models, see Tab. 14) when trained on *perfect* GT data, and classes are known a-priori.

*Table 13.* **Model ablations for data on SSCBench-KITTI360 (Li et al., 2024).** We train the `CAL` model using two different sets of data: pseudo-labels w/o CRF refinement, and with CRF refinement. We report PSC metrics for both variants individually. We observe that training `CAL` with data that has higher label coverage, (pseudo labels w. CRF) clearly improves overall performance except for *stuff* categories.

| Training data for `CAL` | All | | | | Thing | | | Stuff | | | SSC |
|---|---|---|---|---|---|---|---|---|---|---|---|
| Setting | $PQ^\dagger\uparrow$ | PQ↑ | SQ | RQ | PQ | SQ | RQ | PQ | SQ | RQ | mIoU↑ |
| *Pseudo Labels w/o CRF* | | | | | | | | | | | |
| Semantic oracle | 9.85 | 2.96 | 18.41 | 4.77 | 0.06 | 18.42 | 0.11 | 4.42 | 18.40 | 7.11 | 10.15 |
| CLIP semantics | 7.01 | 2.89 | 15.56 | 4.66 | 0.04 | 18.49 | 0.07 | 4.31 | 14.10 | 6.95 | 7.34 |
| *Pseudo Labels w. CRF* | | | | | | | | | | | |
| Semantic oracle | 12.56 | 1.71 | 33.18 | 3.10 | 2.05 | 45.57 | 3.76 | 1.54 | 26.99 | 2.76 | 13.34 |
| CLIP semantics | 8.57 | 1.46 | 21.01 | 2.63 | 1.39 | 27.62 | 2.54 | 1.49 | 17.81 | 2.68 | 8.49 |

*Table 14.* **Panoptic Scene Completion.** On Semantic KITTI (Behley et al., 2019) (val) and SSCBench-KITTI360 (Li et al., 2024) (test). Compared against LMSCNet (Roldao et al., 2020) +MaskPLS (Marcuzzi et al., 2023), JS3CNet (Yan et al., 2021) +MaskPLS (Marcuzzi et al., 2023), SCPNet (Xia et al., 2023) +MaskPLS (Marcuzzi et al., 2023) and PaSCo (Cao et al., 2024) (M=1 and ensemble).

| | Semantic KITTI (Behley et al., 2019) (val set) | | | | | | | | | | | SSCBench-KITTI360 (Li et al., 2024) (test set) | | | | | | | | | | |
| | All | | | | Thing | | | Stuff | | | | All | | | | Thing | | | Stuff | | | |
| Method | $PQ^\dagger\uparrow$ | PQ↑ | SQ | RQ | PQ | SQ | RQ | PQ | SQ | RQ | mIoU↑ | $PQ^\dagger\uparrow$ | PQ↑ | SQ | RQ | PQ | SQ | RQ | PQ | SQ | RQ | mIoU↑ |
|---|---|---|---|---|---|---|---|---|---|---|---|---|---|---|---|---|---|---|---|---|---|---|
| *Fully supervised* | | | | | | | | | | | | | | | | | | | | | | |
| LMSCNet +MaskPLS | 13.81 | 4.17 | 36.13 | 6.82 | 1.62 | 29.87 | 2.68 | 6.02 | 40.69 | 9.82 | 17.02 | 12.76 | 4.14 | 26.52 | 6.45 | 0.88 | 20.41 | 1.58 | 5.78 | 29.58 | 8.88 | 15.10 |
| JS3CNet +MaskPLS | 18.41 | 6.85 | 41.90 | 11.34 | 4.18 | 43.10 | 7.22 | 8.79 | 41.03 | 14.34 | 22.70 | 16.42 | 6.79 | 51.16 | 10.71 | 3.36 | 48.41 | 5.83 | 8.51 | 52.54 | 13.15 | 21.31 |
| SCPNet +MaskPLS | 19.39 | 8.59 | 49.49 | 13.69 | 4.88 | 46.41 | 7.70 | 11.30 | 51.73 | 18.04 | 22.44 | 16.54 | 6.14 | 51.18 | 10.15 | 4.23 | 48.46 | 7.05 | 7.09 | 52.55 | 11.70 | 21.47 |
| PaSCo (M=1) | 26.49 | 15.36 | 54.15 | 23.65 | 12.33 | 47.42 | 18.78 | 17.55 | 59.05 | 27.19 | 28.22 | 19.53 | 9.91 | 58.81 | 15.40 | 3.46 | 57.72 | 6.10 | 13.14 | 59.35 | 20.05 | 21.17 |
| PaSCo (Ensemble) | 31.42 | 16.51 | 54.25 | 25.13 | 13.71 | 48.07 | 20.68 | 18.54 | 58.74 | 28.38 | 30.11 | 26.29 | 10.92 | 56.10 | 17.09 | 4.88 | 57.53 | 8.48 | 13.94 | 55.39 | 21.39 | 22.39 |
| *Pseudo-labels* | | | | | | | | | | | | | | | | | | | | | | |
| CAL (Semantic oracle) | 25.90 | 17.71 | 64.06 | 26.55 | 16.93 | 67.91 | 23.75 | 18.28 | 61.25 | 28.58 | 33.10 | 14.75 | 4.57 | 40.79 | 7.98 | 7.08 | 48.18 | 12.35 | 3.32 | 37.10 | 5.79 | 17.14 |
| CAL (CLIP semantics) | 16.98 | 10.67 | 54.02 | 16.19 | 7.30 | 67.19 | 10.21 | 13.12 | 44.43 | 20.54 | 20.62 | 10.98 | 3.18 | 33.95 | 5.58 | 3.91 | 48.30 | 6.86 | 2.82 | 26.78 | 4.95 | 11.04 |
| *Zero-Shot* | | | | | | | | | | | | | | | | | | | | | | |
| CAL (Semantic oracle) | 17.12 | 6.27 | 43.40 | 10.06 | 3.48 | 44.39 | 5.65 | 8.30 | 42.67 | 13.27 | 20.71 | 12.56 | 1.71 | 33.18 | 3.10 | 2.05 | 45.57 | 3.76 | 1.54 | 26.99 | 2.76 | 13.34 |
| CAL (CLIP semantics) | 13.12 | 5.26 | 27.45 | 8.44 | 2.42 | 22.79 | 3.89 | 7.33 | 30.84 | 11.76 | 13.09 | 8.57 | 1.46 | 21.01 | 2.63 | 1.39 | 27.62 | 2.54 | 1.49 | 17.81 | 2.68 | 8.49 |

**Per-class results.** In Tab. 15, we provide a detailed per-class break-down of the pseudo-label quality and model performance for the SemanticKITTI dataset. We notice that most of the gap between `CAL` and the baselines is due to rare classes (e.g., pedestrian, cyclist), which suggests that fully supervised baselines can exploit class frequency information during training—allowing them to re-balance and weight examples in a way that benefits rare categories. In contrast, because our approach is zero-shot, it does not have access to the ground-truth frequency distributions of classes, and thus it struggles more with less frequent or smaller structures. Additionally, since we rely on K-means for clustering, our method does not incorporate prior knowledge of the data distribution or class frequencies, making it less effective in representing long-tail categories. While this limitation is challenging to address under the zero-shot paradigm, future work could explore more distribution-aware clustering techniques or other strategies that better capture underrepresented object shape priors without relying on labeled data.

## C.4. Additional Qualitative Results

In Fig. 7, we share qualitative results from our model's predictions on the KITTI360 dataset. In Fig. 8, we compare our method `CAL` with zero-shot baselines for the panoptic scene completion task. We also provide a supplementary video visualizing zero-shot panoptic scene completion results for a sequence of Lidar scans.

*Table 15.* Per-class performance analysis for Panoptic Scene Completion, evaluated on SemanticKITTI (Behley et al., 2019) dataset. Per-class scores for the baselines and class-frequencies are taken from (Cao et al., 2024).

| | Method | car (3.92%) | bicycle (0.03%) | motorcycle (0.03%) | truck (0.16%) | other-veh. (0.20%) | person (0.07%) | bicyclist (0.07%) | motorcyclist (0.05%) | road (15.30%) | parking (1.12%) | sidewalk (11.13%) | other-grnd (0.56%) | building (14.10%) | fence (3.90%) | vegetation (39.30%) | trunk (0.51%) | terrain (9.17%) | pole (0.29%) | traf.-sign (0.08%) | mean |
|---|---|---|---|---|---|---|---|---|---|---|---|---|---|---|---|---|---|---|---|---|---|
| **PQ** | LMSCNet + MaskPLS | 9.43 | 0.00 | 0.76 | 2.32 | 0.00 | 0.47 | 0.00 | 0.00 | 53.53 | 1.82 | 5.63 | 0.00 | 0.26 | 0.19 | 0.00 | 0.27 | 3.52 | 1.00 | 0.00 | 4.17 |
| | JS3CNet + MaskPLS | 9.57 | 1.07 | 4.19 | 17.54 | 0.91 | 0.12 | 0.00 | 0.00 | 58.45 | 5.32 | 15.89 | 0.00 | 1.02 | 1.33 | 0.00 | 0.76 | 13.63 | 0.28 | 0.00 | 6.85 |
| | SCPNet + MaskPLS | 18.44 | 4.84 | 6.72 | 4.42 | 2.79 | 1.81 | 0.00 | 0.00 | 63.89 | 7.92 | 19.92 | 0.00 | 3.11 | 3.28 | 0.13 | 2.29 | 21.55 | 1.99 | 0.17 | 8.59 |
| | PaSCo (Ensemble) | 24.55 | 7.82 | 18.09 | 44.89 | 11.32 | 3.00 | 0.00 | 0.00 | 76.22 | 28.12 | 30.42 | 1.33 | 4.85 | 0.27 | 12.97 | 4.22 | 32.61 | 9.69 | 3.26 | 16.51 |
| | **CAL** | 14.11 | 0.00 | 0.00 | 4.34 | 0.88 | 0.00 | 0.00 | 0.00 | 54.08 | 0.00 | 3.09 | 0.00 | 0.12 | 0.00 | 1.00 | 0.00 | 22.20 | 0.14 | 0.00 | 5.26 |
| | **CAL** - Pseudo Labels | 20.24 | 1.38 | 7.29 | 8.88 | 5.29 | 1.06 | 13.03 | 1.21 | 62.60 | 1.93 | 8.50 | 0.00 | 12.48 | 0.00 | 33.55 | 0.00 | 23.20 | 1.04 | 1.22 | 10.67 |
| **SQ** | LMSCNet + MaskPLS | 62.65 | 0.00 | 53.44 | 53.87 | 0.00 | 69.00 | 0.00 | 0.00 | 63.30 | 57.83 | 52.70 | 0.00 | 53.93 | 52.58 | 0.00 | 59.76 | 54.12 | 53.37 | 0.00 | 36.13 |
| | JS3CNet + MaskPLS | 59.88 | 53.79 | 55.17 | 57.73 | 55.70 | 62.50 | 0.00 | 0.00 | 65.98 | 55.70 | 54.53 | 0.00 | 52.62 | 53.41 | 0.00 | 55.01 | 56.38 | 57.68 | 0.00 | 41.90 |
| | SCPNet + MaskPLS | 66.69 | 57.78 | 65.30 | 55.30 | 65.15 | 61.01 | 0.00 | 0.00 | 68.56 | 58.72 | 55.81 | 0.00 | 54.94 | 54.45 | 51.04 | 55.58 | 59.86 | 52.97 | 57.14 | 49.49 |
| | PaSCo (Ensemble) | 70.10 | 57.84 | 67.00 | 67.33 | 62.15 | 60.14 | 0.00 | 0.00 | 77.52 | 62.62 | 59.95 | 54.71 | 55.87 | 51.29 | 52.85 | 57.50 | 63.88 | 54.78 | 55.17 | 54.25 |
| | **CAL** | 65.84 | 0.00 | 0.00 | 52.80 | 63.71 | 0.00 | 0.00 | 0.00 | 63.05 | 0.00 | 54.81 | 0.00 | 50.02 | 0.00 | 51.47 | 0.00 | 62.58 | 57.32 | 0.00 | 27.45 |
| | **CAL** - Pseudo Labels | 74.38 | 54.58 | 67.01 | 74.56 | 74.12 | 64.88 | 71.65 | 56.35 | 69.14 | 58.48 | 58.24 | 0.00 | 56.52 | 0.00 | 57.95 | 0.00 | 68.15 | 60.64 | 59.67 | 54.02 |
| **RQ** | LMSCNet + MaskPLS | 15.05 | 0.00 | 1.42 | 4.30 | 0.00 | 0.67 | 0.00 | 0.00 | 84.56 | 3.15 | 10.69 | 0.00 | 0.48 | 0.37 | 0.00 | 0.45 | 6.50 | 1.87 | 0.00 | 6.82 |
| | JS3CNet + MaskPLS | 15.98 | 2.00 | 7.59 | 30.38 | 1.63 | 0.20 | 0.00 | 0.00 | 88.59 | 9.55 | 29.14 | 0.00 | 1.93 | 2.49 | 0.00 | 1.39 | 24.17 | 0.48 | 0.00 | 11.34 |
| | SCPNet + MaskPLS | 27.65 | 8.38 | 10.29 | 8.00 | 4.28 | 2.96 | 0.00 | 0.00 | 93.18 | 13.50 | 35.69 | 0.00 | 5.66 | 6.03 | 0.25 | 4.12 | 36.00 | 3.76 | 0.30 | 13.69 |
| | PaSCo (Ensemble) | 35.03 | 13.51 | 27.00 | 66.67 | 18.21 | 4.98 | 0.00 | 0.00 | 98.32 | 44.91 | 50.73 | 2.44 | 8.69 | 0.52 | 24.54 | 7.33 | 51.05 | 17.70 | 5.91 | 25.13 |
| | **CAL** | 21.52 | 0.00 | 0.00 | 8.22 | 1.39 | 0.00 | 0.00 | 0.00 | 85.77 | 0.00 | 5.64 | 0.00 | 0.24 | 0.00 | 1.94 | 0.00 | 35.47 | 0.24 | 0.00 | 8.44 |
| | **CAL** - Pseudo Labels | 27.21 | 2.54 | 10.88 | 11.91 | 7.14 | 1.64 | 18.18 | 2.15 | 90.54 | 3.31 | 14.60 | 0.00 | 22.08 | 0.00 | 57.89 | 0.00 | 33.75 | 1.72 | 2.04 | 16.19 |

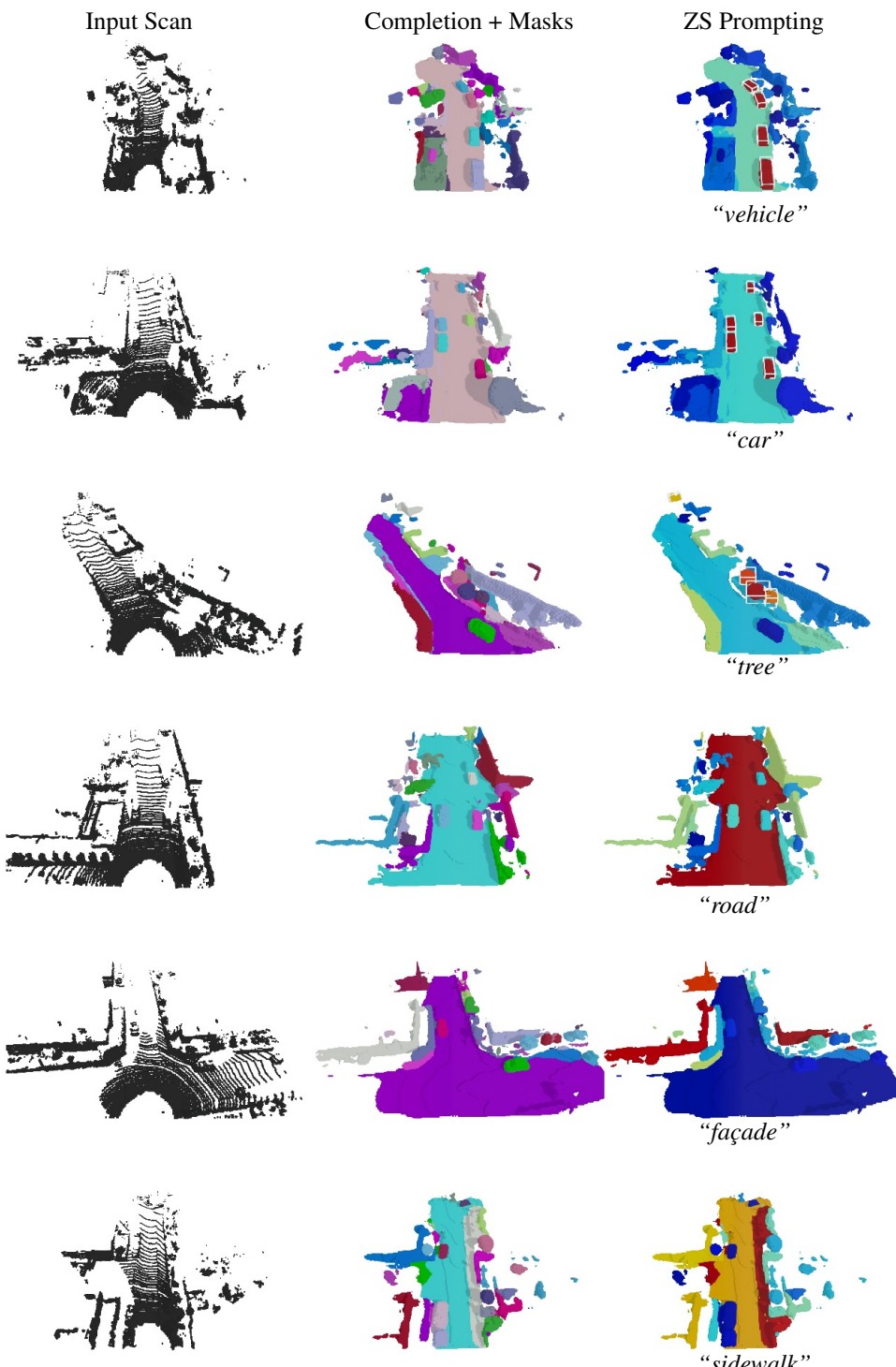

*Figure 7.* **Qualitative results on KITTI-360 (**Liao et al., 2021**).** Given a single Lidar scan as input ($1^{st}$ column), **CAL** completes object-level observations as a set of masks over the voxel grid ($2^{nd}$ *column*) with semantic CLIP feature for each predicted mask. We can prompt with any semantic class vocabulary and perform panoptic and semantic scene completion ($3^{rd}$ column).

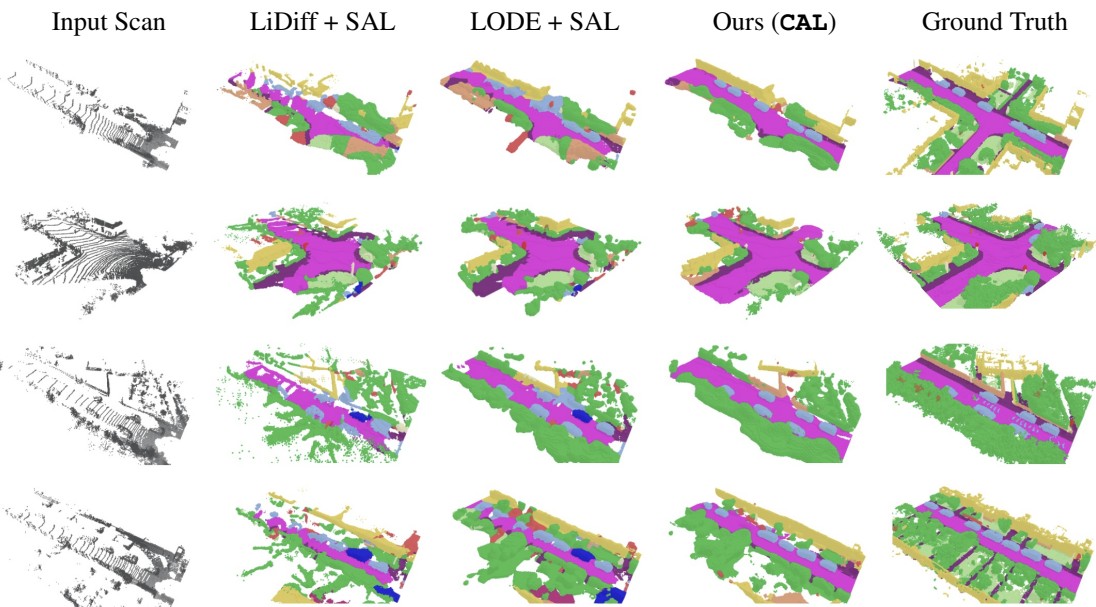

*Figure 8.* **Qualitative comparison to zero-shot baselines on SemanticKITTI.** Given a single Lidar scan ($1^{st}$ col.), we compare our method (**CAL**, $4^{th}$ col.) to zero-shot baselines ($2^{nd}$ and $3^{rd}$ cols.) combining LiDiff (Nunes et al., 2024) and LODE (Li et al., 2023b) with SAL (Osep et al., 2024). While baselines struggle with coherent structure and semantic accuracy, CAL produces cleaner and more complete outputs that align closely with the ground truth.

