# OpenReview forum: "Towards Learning to Complete Anything in Lidar"
_ICML.cc/2025/Conference — ICML 2025 poster_

### Official Review · Reviewer_ptaj · 2025-03-13

**Overall Recommendation:** 3

**Summary:**

The paper proposes a zero-short learning method CAL (Complete Anything in Lidar) to use the temporal context from multi-modal sensor sequences to mine object shapes and semantic features that are then distilled into a Lidar-only instance-level completion and recognition model. The experiments on real-world lidar benchmarks demonstrate that the approach can do zero-shot shape completion with promising results.

**Claims And Evidence:**

Yes, the claim “CAL performs zero-shot PSC in lidar ” is shown by quantitative experiments on two established datasets with comparions on zero-shot performance to baselines. Furthermore, the paper also shows qualitative results on the recognition of unlabeled classes.

**Essential References Not Discussed:**

None

**Experimental Designs Or Analyses:**

The ablation studies are thorough, showing the impact of different design and pseudo-labeling choices. However, the pseudo-labeling approach is complicated, which may introduce errors, which makes the whole pipeline difficult to train. Hence, it would be good to explain this more to make the paper more solid and convincing.

**Methods And Evaluation Criteria:**

Yes, the panoptic scene completion metrics, such as PQ, SQ, RQ, and mIoU are standard and appropriate for existing benchmarks
 including SemanticKITTI and KITTI-360.

**Other Comments Or Suggestions:**

None

**Other Strengths And Weaknesses:**

Weakness
1. The method heavily relies on the quality of 2D foundation models and multi-frame projection. Even though CRF-based refinements can compensate for partial or noisy coverage, the paper acknowledges that coverage remains imperfect.
2. The whole pipeline looks computationally heavy: pseudo-labels are built using multi-frame and multi-sensor pipelines, which might be acceptable for offline training but limits applications on real-time scenarios.

**Questions For Authors:**

1. How good is pseudo-label accuracy in practice? It would be good to provide some analysis of where the pseudo-label pipeline fails.
2. How sensitive is the approach to inaccuracies or biases from the 2D foundation models?

**Relation To Broader Scientific Literature:**

The paper is directly related to unsupervised learning and semi-supervised learning. Moreover, the paper is also related to large vision language models, which is helpful for multi-modal learning.

**Theoretical Claims:**

The paper does not contain complex proofs or new theoretical analysis, and is more on experimental results. The pipeline looks standard and well-documented.

---

> ### Author Rebuttal · Authors · 2025-03-31
>
> We thank the reviewer for their feedback. We are happy to hear that the reviewer found that our experiments demonstrate promising results for zero-shot shape completion. We are also glad that the reviewer has found our ablation studies thorough, and our method well-documented. Below, we address the comments and questions posed by the reviewer.
>
> **Q1. How good is pseudo-label accuracy in practice and when does it fail?**
>
> This is a great question, and we acknowledge its importance. In the main paper, we carefully investigated pseudo-label accuracy through extensive experiments and discussions. As reported in *Section 4.3* and *Appendix C.2*, our experiments indicate strong overall pseudo-labeling accuracy. The accuracy of pseudo-labels primarily depends on recognizing and completing objects by tracking them across video frames, integrating multi-view observations, and leveraging zero-shot recognition via vision-foundation model features. In the following, we report the main findings from our experiments:
>
> - *Table 2* investigates pseudo-label accuracy when varying the number of tracked frames ($T_{fw}$ and $T_{bw}$), and the tracking stride $w$. Our experiments suggest that a sufficient number of frames is essential for completing objects based on observations across multiple views. Our model is generally robust to view changes, and CRF refinement allows to reduce the number of tracked frames (*Table 10*). We notice that failure cases primarily stem from the foundation model incorrectly switching tracking IDs (in case of strong view changes or occlusions), or failing to recognize objects, highlighting areas for further improvement.
> - *Table 3* evaluates pseudo-label accuracy by measuring coverage completeness against GT annotations. While pseudo-labels may initially provide partial scene completions due to tracking limitations, our CRF refinement significantly enhances label coverage. However, challenges persist when objects are completely undetected or are located in poorly visible regions.
> - *Table 4* evaluates pseudo-label quality on two datasets for both semantic-oracle and zero-shot settings (see last two rows). We notice that our pseudo-labels demonstrate strong performance even under zero-shot conditions, while revealing room for improvement in terms of the quality of vision-foundation model features for zero-shot recognition.
>
> **Q2. How sensitive is the approach to inaccuracies or biases from the 2D foundation models?**
>
> Our method is generally robust to minor errors from video foundation models like SAM2, thanks to both our pseudo-labeler and our training strategy. The pseudo-labeler enhances robustness by aggregating labels across frames, refining 3D masks per scan, and applying CRF refinement to improve label coverage. On the training side, we employ losses and training task formulations (see *Table 5*) that are specifically designed to help the model learn effectively from potentially-noisy pseudo-labels, ensuring it remains resilient even when pseudo-label coverage is imperfect. However, inaccuracies can still arise, particularly from 2D mask tracking failures, such as ID switches during significant view changes or heavy occlusions. To minimize these issues, our pipeline selectively generates pseudo-labels only for objects with high-confidence completions, filtering out lower-confidence outputs. Additionally, we also fine-tuned video-based tracking parameters (as shown in *Table 2*) to further reduce errors.  Importantly, our pseudo-labeling engine is modular, meaning that one can easily integrate improved 2D foundation models as they become available, which may translate to direct enhancements in pseudo-labeling performance.
>
> **Q3. The proposed pseudo-labeling approach is computationally heavy, which may limit applications to real-time scenarios.**
>
> While we acknowledge that our pseudo-labeling engine is computationally intensive, we optimize this by performing pseudo-labeling offline to generate a training dataset, which is then distilled into a single, efficient model during training. Once this model is trained, inference (i.e., completing a sparse LiDAR scan) only requires a single forward pass through the model, making it more suitable for real-time scenarios. Although real-time performance is not the main focus of this work, future efforts could improve the efficiency of our pseudo-labeler to further improve scalability.
>
> In this direction, we note that the primary computational cost arises from the video foundation model used for object tracking across an RGB sequence. To improve efficiency, one may consider reducing the number of frames used during 2D mask propagation. Our preliminary results reported in *Table 10* suggest that we can (significantly) reduce the tracking horizon and minimize computational costs while achieving comparable performance.

---

### Official Review · Reviewer_C64v · 2025-03-14

**Overall Recommendation:** 3

**Summary:**

The paper introduces CAL (Complete Anything in Lidar), a zero-shot panoptic scene completion framework that infers dense 3D object and scene geometry from sparse Lidar scans without relying on predefined class vocabularies. To achieve this, the authors propose a pseudo-labeling engine that mines 3D shape priors from unlabeled Lidar sequences by leveraging vision foundation models for object segmentation and tracking in videos. These mined pseudo-labels, which combine shape completion and semantic features, are then used to train CAL, a sparse generative encoder-decoder network with a transformer-based instance decoder that performs class-agnostic segmentation and completion. Unlike prior methods, CAL enables zero-shot semantic and panoptic scene completion, amodal 3D object detection, and recognition of novel object classes at test time via text-based prompting. The experiments on SemanticKITTI and SSCBench-KITTI360 show that CAL does not match the performance of fully supervised baselines.

**Claims And Evidence:**

Yes.

**Essential References Not Discussed:**

In LiDAR-based segmentation, numerous semi-supervised or weakly supervised methods [1, 2], which do not fully rely on (fully-) manually labeled datasets, have not been discussed. Additionally, related works [3] utilizing vision foundation models have also not been discussed. The authors should explicitly compare their method against these relevant methods.


[1] Li Li, Hubert P. H. Shum, Toby P. Breckon; Proceedings of the IEEE/CVF Conference on Computer Vision and Pattern Recognition (CVPR), 2023, pp. 9361-9371

[2] Ozan Unal, Dengxin Dai, Luc Van Gool; Proceedings of the IEEE/CVF Conference on Computer Vision and Pattern Recognition (CVPR), 2022, pp. 2697-2707

[3] Liu, Youquan, et al. "Segment any point cloud sequences by distilling vision foundation models." Advances in Neural Information Processing Systems 36 (2023): 37193-37229.

**Experimental Designs Or Analyses:**

Checked. See Other Strengths And Weaknesses

**Methods And Evaluation Criteria:**

Checked. See Other Strengths And Weaknesses

**Other Comments Or Suggestions:**

My concerns mainly focus on the performance (see Other Strengths And Weaknesses) and novelty (see Relation To Broader Scientific Literature) of the proposed method. If the authors can adequately address these concerns, I am willing to increase my rating.

**Other Strengths And Weaknesses:**

Strengths:
- The paper introduces CAL (Complete Anything in Lidar), a novel zero-shot panoptic scene completion approach. It extends beyond traditional fixed taxonomies by learning object shape priors from unlabeled temporal Lidar sequences.
- The paper is well-structured, with clear motivation and methodology.

Weakness:
- The method is compared to fully supervised baselines (Tab. 1), but an ablation against other zero-shot methods would strengthen the effectiveness. Although I acknowledge that the authors claim this is the first method for zero-shot panoptic scene completion in LiDAR, and therefore it might be difficult to find a second existing zero-shot method for direct comparison, perhaps a practical alternative could be modifying current supervised methods (e.g., by adding specific modules or other adjustments) to adapt them for zero-shot evaluation.
- It is important to point out that there remains a substantial performance gap (Tab. 1) between existing zero-shot methods and supervised approaches. Consequently, I am not convinced that the zero-shot method proposed in this paper could be effectively applied to the application scenarios mentioned in Fig. 1.
- The reliance on pre-trained 2D models (CLIP, SAM) may inherit their biases and limitations.

**Questions For Authors:**

N/A

**Relation To Broader Scientific Literature:**

The idea of distilling vision foundation models (VFMs) into LiDAR-specific models is no longer novel in the broader literature; for example, prior works have successfully applied VFMs to LiDAR panoptic segmentation [1] and semantic segmentation [2] tasks. This paper extends a similar method to a different yet related task—specifically, instance-level completion and recognition using LiDAR data alone. Thus, the key contribution here builds upon existing insights, following established approaches from other LiDAR VFM-based tasks to instance-level scene completion and recognition.

[1] Osep, A., Meinhardt, T., Ferroni, F., Peri, N., Ramanan, D.,and Leal-Taixe, L. Better call sal: Towards learning to segment anything in lidar. In Eur. Conf. Comput. Vis., 2024.
[2] Liu, Youquan, et al. "Segment any point cloud sequences by distilling vision foundation models." Advances in Neural Information Processing Systems 36 (2023): 37193-37229.

**Theoretical Claims:**

Checked.

---

> ### Author Rebuttal · Authors · 2025-03-31
>
> We’re delighted that the reviewer found our paper well-structured with clear motivation and methodology. We appreciate the detailed feedback, and we are excited to address the concerns raised by the reviewer.
>
> **Q1. Novelty of distilling vision foundation models (VFMs) to Lidar**
>
> We agree with the reviewer that distilling VFMs to Lidar is not entirely novel in the broader literature; for instance, [1,2] explore this for *segmentation*. Our work, however, introduces zero-shot Lidar *panoptic completion*, which extends beyond segmentation. While we leverage prior insights on distilling CLIP features for zero-shot (ZS) prompting—as in [1]—we find this alone insufficient for *completing* 3D shapes from sparse LiDAR scans. To this end, we combine ZS prompting with temporal aggregation of objects, providing essential cues for shape completion. Generating supervision for ZS panoptic completion presents non-trivial challenges: such as associating objects across time, partial coverage and occlusions, and training models to reconstruct full shapes from incomplete labels. These unique complexities in *completion* distinguish our method from prior work on ZS Lidar *segmentation* such as [1]. We believe that, as *Reviewer KxNA* also noted, our work tackles a *“novel and underexplored problem”* with *“significant value for the research community”*.
>
> **Q2. Comparison to zero-shot (ZS) baselines**
>
> We thank the reviewer for the suggestion, and agree that ZS baseline comparisons would strengthen our analysis. As the reviewer noted, our method is the *“first for ZS panoptic scene completion in LiDAR”*— making it *“difficult to find a second existing ZS method for direct comparison”*.
>
> To this end, we constructed two baselines adhering to the following criteria for a fair ZS comparison: (1) input is a single Lidar scan, (2) scene completion model is trained *without semantic labels*, and (3) instance prediction and semantic inference rely on *zero-shot* recognition. Accordingly, we combined recent Lidar completion methods w/o semantic labels—LODE [5] and LiDiff [6]—with SAL [1], a ZS panoptic segmentation method. As SAL’s codebase is not public, we obtained its ZS predictions on SemanticKITTI directly from its authors. Our baselines are:
>
> - LODE + SAL: LODE [5] performs implicit scene completion from sparse LiDAR, trained with GT completion but no sem. labels. We extract a surface mesh from its output, convert it to an occupancy grid, and propagate SAL’s ZS panoptic labels to voxels.
> - LiDiff + SAL: LiDiff [6], a diffusion-based completion method using GT completion data (no sem. labels), densifies LiDAR point clouds. We convert its output to an occupancy grid and similarly propagate SAL’s ZS panoptic labels to occupied voxels.
>
> |                            | All PQ† | All PQ | All SQ | All RQ | Thing PQ | Thing SQ | Thing RQ | Stuff PQ | Stuff SQ | Stuff RQ | mIoU  |
> |-|-|-|-|-|-|-|-|-|-|-|-|
> | LODE + SAL  | 7.74    | 1.96   | 11.12  | 3.54   | 0.00     | 6.36     | 0.00     | 3.39     | 14.59    | 6.11     | 8.12  |
> | LiDiff + SAL  | 7.35    | 0.36   | 23.95  | 0.65   | 0.22     | **34.81**    | 0.40     | 0.46     | 16.06    | 0.83     | 7.38  |
> | **Ours**              | **13.12** | **5.26** | **27.45** | **8.44** | **2.42** | 22.79    | **3.89** | **7.33** | **30.84** | **11.76** | **13.09** |
>
> Results show our method outperforms the baselines across nearly all metrics. Notably, while the baselines leverage completion models trained on fully completed GT, our approach excels despite using pseudo-labels with only partial coverage. This highlights that ZS panoptic Lidar scene completion is a challenging task, not trivially solved by existing methods.
>
> **Q3. Performance gap between fully-supervised and zero-shot methods**
>
> Due to space limits, please see our response to *Reviewer KxNA (Q2)*.
>
> **Q4. VFM biases and limitations**
>
> Due to space limits, please see our responses to *Reviewers KxNA (Q1) and ptaj (Q1-2)*.
>
> **Q5. Additional references**
>
> Thanks for the valuable references! We'll add them to the Lidar-based *segmentation* section. [3] and [4] explore weakly supervised segmentation, while [2] uses contrastive pre-training to distill VFMs for segmentation. As the reviewer noted, [2–4] reduce manual labeling efforts. In contrast, our method addresses ZS Lidar panoptic *completion* with distinct challenges beyond the segmentation scope [2–4].
>
> [1] Osep et al., Better Call SAL: Towards Learning to Segment Anything in Lidar, ECCV '24
>
> [2] Liu et al., Segment Any Point Cloud Sequences by Distilling Vision Foundation Models, NeurIPS '23
>
> [3] Li et al., Less is More: Reducing Task and Model Complexity for 3D Point Cloud Semantic Segmentation, CVPR '23
>
> [4] Unal et al., Scribble-Supervised LiDAR Semantic Segmentation, CVPR '22
>
> [5] Li et al., LODE: Locally Conditioned Eikonal Implicit Scene Completion from Sparse LiDAR, ICRA '23
>
> [6] Nunes et al., Scaling Diffusion Models to Real-World 3D LiDAR Scene Completion, CVPR '24

---

> > ### Comment · Reviewer_C64v · 2025-04-07
> >
> > Thank you for the rebuttal. It has addressed most of my concerns. I hope the authors can attach some of the experiments and discussions in their revised manuscripts and supplementary material.
> >
> > I've updated my rating to WEAK ACCEPT.

---

> > > ### Author Response · Authors · 2025-04-08
> > >
> > > Dear Reviewer C64v,
> > >
> > > We are truly grateful for your support and updated score - we're very glad to hear that we were able to address your concerns! Thank you once again for your thoughtful feedback - we will incorporate these additional experimental findings as well as discussions in the revised version of our paper.
> > >
> > > Best regards,
> > >
> > > Authors

---

### Official Review · Reviewer_KxNA · 2025-03-15

**Overall Recommendation:** 3

**Summary:**

This paper introduces a novel zero-shot approach for completing data from a single LiDAR scan, including both object and instance completion. The method is potentially scalable as it leverages a pre-trained foundational video segmentation model, eliminating the need for labeled video data. CLIP features are extracted and fused across multiple views to supervise the completion model. The extensive experimental results, along with the implementations and discussions provided in the supplementary materials, offer valuable insights into the system’s design. While the results do not outperform fully supervised methods, the proposed approach holds significant value for the research community due to its scalability.

**Claims And Evidence:**

The paper addresses a novel and underexplored problem: completing parts from sparse lidar inputs. The motivation for the study is well-articulated, and the research problem itself holds significant value.

Addressing the completion problem in an open-vocabulary setting is highly valuable, as it presents intriguing possibilities for real-world applications.

The authors claim to present the first method for Zero-Shot Panoptic Scene Completion using LiDAR data, which adds a unique contribution to the field.

**Essential References Not Discussed:**

N/A

**Experimental Designs Or Analyses:**

The experiments are quite extensive and provide comparisons with various fully supervised baselines. However, Table 1 shows a noticeable gap in performance compared to these methods. Could you provide insights into the main reasons behind this gap and suggest ways to improve the performance further? Although the authors claim that the main gap arises from rare classes, the ratio of data for these rare classes does not fully explain the significant performance gap.

**Methods And Evaluation Criteria:**

Using a video foundation model to extract masks for associating temporal information, and then extracting and aggregating CLIP features in 3D space, is a sound and reasonable approach. However, reliance on the video foundation model, which may not always be perfectly accurate, could introduce errors and potentially limit the overall performance of the system.


The title suggests that the method completes objects solely from LiDAR data, while the approach described in the paper still relies on camera images. This discrepancy could be misleading.

The figures do not clearly illustrate how the CLIP features are fused together.

**Other Comments Or Suggestions:**

N/A

**Other Strengths And Weaknesses:**

Completing data from sparse views is a highly valuable problem, as it provides fundamental information for downstream tasks such as object detection, as demonstrated by the authors in the paper.

**Questions For Authors:**

How does the reliance on the foundational video model potentially affect the final performance, and what steps can be taken to mitigate any errors introduced by it?

What are the main reasons behind the performance gap observed in Table 1 when comparing your method to fully supervised baselines?

**Relation To Broader Scientific Literature:**

N/A

**Theoretical Claims:**

The methods presented follow a typical learning-based formulation, and therefore, extensive theoretical proofs are not required. The foundational theories related to occupancy networks, LiDAR/camera geometry, and the reliance on the video foundation model are sound and reasonable.

---

> ### Author Rebuttal · Authors · 2025-03-31
>
> We are thrilled that the reviewer finds our task of zero-shot Lidar-based panoptic scene completion challenging and novel. We are particularly happy that the reviewer recognizes our method's scalability potential (wrt. data) and appreciates the extensive experimental results and sound methodology. Below, we address questions raised by the reviewer.
>
>
> **Q1. The reviewer asks about the effect of the video-foundation model on the final performance and asks how errors it introduces can be mitigated.**
>
> Great point; indeed, predictions from the video foundation model may not always be perfectly accurate, as noted by the reviewer. In practice, we observed two types of errors: (i) noisy masks, especially inaccurate around borders, that cause artifacts during image-to-Lidar lifting. To address these, we perform several post-processing steps described in the paper, most importantly, DBSCAN-based segment refinement and CRF-based refinement of aggregated labels. (ii) video-segmentation models may also produce tracking errors, such as ID switches. To address this, we empirically determined the window size and stride in which state-of-the-art models are reliable (which we also ablated in *Table 2* of the paper), ensuring that such errors are infrequent. In practice, we find that as long as we have a sufficiently high signal-to-noise ratio, our model learns to ignore such artifacts. To mitigate these errors in future work, one could additionally utilize 3D/4D geometric cues from the Lidar sequence to improve temporal association.
>
> **Q2. Performance gap between fully-supervised and zero-shot methods.**
>
> Great question! We would like to elaborate further on the main reasons behind this gap (beyond the limitations due to rare classes). As shown by our findings in *Table 1* (semantic oracle vs. zero-shot results), this gap is largely due to zero-shot semantic recognition— which is a challenging and active research area. Several opportunities exist to improve performance, such as enhancing the underlying Vision Language Model (VLM) for zero-shot recognition, or incorporating manually labeled data for supervised fine-tuning.
>
> Another potential cause for this gap is that our pseudo-labels have lower coverage (approximately 50% on KITTI-360 and 70% on SemanticKITTI, please refer to *Table 3*) compared to ground truth labels, due to their construction requiring camera-Lidar co-visibility. In contrast, fully supervised methods benefit from training on a complete ground-truth signal with full-grid coverage (see *Figure 4*, 4$^{th}$ column), effectively using more labeled data. While this lower coverage is a limitation when training on fixed-size datasets like SemanticKITTI, as the *Reviewer KxNA* notes, it also underscores our approach’s potential: scaling with more data could help close the gap with fully supervised, closed vocabulary baselines. Our method can already be applied to the application scenarios from *Fig. 1* (as also shown by quantitative ZS-panoptic completion results and qualitative results). However, scaling with more data could enable our method to be more effective and robust for these tasks in the future.
>
> On a related note, following *Reviewer C64v*’s suggestion, we added zero-shot panoptic scene completion baselines (please refer to our response to *Reviewer C64v*, *Q2*), which further confirm that zero-shot Lidar panoptic scene completion is a problem with non-trivial challenges.
>
> We appreciate *Reviewer KxNA*'s interpretation that our method *“holds significant value for the research community even though it does not necessarily outperform the fully supervised methods”*. We will ensure to include this extended discussion in the paper to further elaborate on the gap between our zero-shot method and fully supervised baselines, as well as potential solutions to address this gap.
>
> **Q3. The reviewer highlights that Figure 2 does not clearly illustrate how the CLIP features are fused.**
>
> Thanks for pointing this out! We compute CLIP features per-instance in every frame and *average* these (normalized) CLIP features over time across the frames in which this instance was observed and tracked. We will improve the clarity of this figure to highlight this temporal aggregation step, and also expand our textual description in L198.
>
> **Q4. Reviewer notes that our title may be misleading as the pseudo-labeling engine in our approach still relies on camera images.**
>
> Thank you for raising this point! Our trained (distilled) model indeed takes only a single Lidar scan as input at test time, producing a completed scene representation without using any camera data during inference. However, as the reviewer points out, our pseudo-labeling engine—used solely during training—leverages camera images and Lidar sequences to generate training labels. To avoid any confusion, we’re happy to revise the title (for example, a possible alternative could be "Towards Learning to Complete Anything in Lidar Using Vision Foundation Models").

---

### Decision · Program_Chairs · 2025-05-01

**Decision:**

Accept (poster)

**Comment:**

The paper received three "weak accept" ratings after the rebuttal. Two of the reviewers maintained their initially positive ratings.
The reviewer who initially gave a "weak reject" rating later changed it to a "weak accept," as the authors satisfactorily addressed the concerns regarding performance and novelty.
The area chair agrees with the consensus of the reviewers and therefore recommends a "weak accept" if there is room in the program.